# Sublithospheric diamond ages and the supercontinent cycle

Suzette Timmerman[1,15 ✉], Thomas Stachel[1], Janne M. Koornneef[2], Karen V. Smit[3], Rikke Harlou[4], Geoff M. Nowell[4], Andrew R. Thomson[5], Simon C. Kohn[6], Joshua H. F. L. Davies[7], Gareth R. Davies[2], Mandy Y. Krebs[1], Qiwei Zhang[1], Sarah E. M. Milne[1], Jeffrey W. Harris[8], Felix Kaminsky[9], Dmitry Zedgenizov[10], Galina Bulanova[6], Chris B. Smith[6], Izaac Cabral Neto[11], Francisco V. Silveira[11], Antony D. Burnham[12], Fabrizio Nestola[13], Steven B. Shirey[14], Michael J. Walter[14], Andrew Steele[14] & D. Graham Pearson[1]

Subduction related to the ancient supercontinent cycle is poorly constrained by mantle samples. Sublithospheric diamond crystallization records the release of melts from subducting oceanic lithosphere at 300–700 km depths[1,2] and is especially suited to tracking the timing and effects of deep mantle processes on supercontinents. Here we show that four isotope systems (Rb–Sr, Sm–Nd, U–Pb and Re–Os) applied to Fe-sulfide and $CaSiO_3$ inclusions within 13 sublithospheric diamonds from Juína (Brazil) and Kankan (Guinea) give broadly overlapping crystallization ages from around 450 to 650 million years ago. The intracratonic location of the diamond deposits on Gondwana and the ages, initial isotopic ratios, and trace element content of the inclusions indicate formation from a peri-Gondwanan subduction system. Preservation of these Neoproterozoic–Palaeozoic sublithospheric diamonds beneath Gondwana until its Cretaceous breakup, coupled with majorite geobarometry[3,4], suggests that they accreted to and were retained in the lithospheric keel for more than 300 Myr during supercontinent migration. We propose that this process of lithosphere growth—with diamonds attached to the supercontinent keel by the diapiric uprise of depleted buoyant material and pieces of slab crust—could have enhanced supercontinent stability.

Earth's supercontinent cycle, driven by plate tectonics, results in large-scale episodic mixing of the deep mantle due to subduction along supercontinent margins, sometimes localized into specific hemispheres[5]. The record of this subduction is poorly defined and mostly based on surface information such as the composition of crustal rocks and palaeomagnetism. Sublithospheric diamonds from Juína (Brazil) and Kankan (Guinea, West Africa), erupted through palaeo-Gondwana lithosphere by kimberlites in the Cretaceous, offer a new, deeply derived diamond perspective on the timing and nature of such subduction processes, including the modification of the Gondwana mantle keel (Extended Data Fig. 1).

Sublithospheric diamonds come from depths of 300 km to more than 700 km (refs. 1,6) and provide the deepest available samples of Earth's mantle. The chemical signatures of most sublithospheric diamonds and their inclusions reflect subducted oceanic protoliths[2], with carbon derived from recycled crustal carbonates and/or organic carbon in altered oceanic crust[7,8]. Silicate inclusions hosted in diamonds document interaction of former oceanic crust with Earth's hydrosphere[9,10]. Despite this, sublithospheric diamond ages are poorly constrained, hindering a broader geological understanding of their chemical signatures. Limited previous attempts to date sublithospheric diamonds produced no isochronous relationships but hinted at younger ages than most lithospheric diamonds, and a link to subduction beneath Gondwana[11,12]. Here we use new age data to document the relationships between peri-Gondwanan subduction, diamond formation and the subsequent evolution of this supercontinent.

Sublithospheric diamonds crystallize from a variety of fluids, such as carbonated or hydrous melts, methane-rich fluids/melts or metallic melts[3,13]. Mineral assemblages, major and trace element compositions, and stable isotope compositions show that the sublithospheric diamonds studied here precipitated during redox-freezing events[14] when slab-derived carbonated melts reacted with the surrounding mantle at depths of 300 km to more than 700 km (refs. 15,16) or in a redox-melting process also involving subducted material[17]. Diamonds

[1]Department of Earth and Atmospheric Sciences, University of Alberta, Edmonton, Alberta, Canada. [2]Faculty of Sciences, Vrije Universiteit, Amsterdam, The Netherlands. [3]School of Geosciences, University of Witwatersrand, Johannesburg, South Africa. [4]Department of Earth Sciences, University of Durham, Durham, UK. [5]Department of Earth Sciences, University College London, London, UK. [6]School of Earth Sciences, University of Bristol, Bristol, UK. [7]Département des sciences de la Terre et de l'atmosphère, Université du Québec à Montréal, Montreal, Quebec, Canada. [8]School of Geographical and Earth Sciences, University of Glasgow, Glasgow, UK. [9]V. I. Vernadsky Institute of Geochemistry and Analytical Chemistry, Russian Academy of Sciences, Moscow, Russian Federation. [10]A. N. Zavaritsky Institute of Geochemistry, Russian Academy of Sciences, Ekaterinburg, Russian Federation. [11]CPRM/SGB, Geological Survey of Brazil, Natal, Brazil. [12]Research School of Earth Sciences, Australian National University, Canberra, Australian Capital Territory, Australia. [13]Department of Geosciences, University of Padua, Padua, Italy. [14]Earth and Planets Laboratory, Carnegie Institution for Science, Washington, DC, USA. [15]Present address: Institute for Geological Sciences, University of Bern, Bern, Switzerland. ✉e-mail: suzette.timmerman@geo.unibe.ch

(Extended Data Fig. 2) were selected based on the presence of Ca-silicate inclusions (breyite, larnite or $CaSi_2O_5$-titanite; Extended Data Fig. 3), many of which have been interpreted as former Ca-silicate perovskite[3]. The diamonds are from the Juína area, Brazil ($n = 10$: Rio Sorriso and Massinha alluvials, Juína-5 kimberlite and Collier-4 kimberlite) and from Kankan, Guinea ($n = 2$, both alluvials). Fourteen macro-Ca-silicate inclusions and two micro-mineral/fluid inclusion-bearing areas within diamonds were analysed for trace elements and isotope systematics (Methods). Additionally, a sulfide in diamond J1 from the Collier-4 kimberlite was selected (Extended Data Figs. 4–7).

All studied diamonds show characteristics typical of a sublithospheric origin, either containing no detectable nitrogen by Fourier transform infrared (FTIR) spectroscopy (Type II; Supplementary Tables 1 and 2) or low nitrogen contents that are fully aggregated (100% of nitrogen as $N_4V-B$ centres). Kankan diamonds (KK99, KK200) contain breyite, larnite and coesite inclusions, among other phases (Supplementary Table 1 (refs. 16,18,19)). Growth zones adjacent to the analysed Ca-silicate inclusions in Kankan diamonds have positive $\delta^{13}C$ values[16], typical of subducted marine carbonate. The studied diamonds from the Juína region contain breyite ($CaSiO_3$) inclusions, sometimes in association with larnite ($Ca_2SiO_4$), and $CaSi_2O_5$-titanite. Half of these diamonds ($n = 5$) contain ferropericlase inclusions ((Mg,Fe)O) that could form at lower mantle pressure[1,6], but can also form when carbonated melts react with reduced mantle peridotite at deep upper mantle and transition zone pressures[20,21]. Two of the Juína diamonds were previously determined to have a metabasaltic source, indicated by elevated $\delta^{18}O$ values of an associated majoritic garnet inclusion[9,15]. A Massinha (Juína) diamond contains breyite inclusions as well as a composite inclusion of diopside + merwinite + calcite + olivine (molar Mg/(Mg + Fe) around 88; Supplementary Table 3). This assemblage could represent the incomplete reaction of calcite + olivine → merwinite ($Ca_3Mg[SiO_4]_2$), in the presence of pyroxene, indicating slab-derived Ca-carbonated melt interacting with peridotite[22]. Lanthanum concentrations of the Ca-silicate inclusions vary up to 10,000 times CI-chondrite levels (Extended Data Fig. 8 and Supplementary Table 4) and many inclusions have chondrite-normalized light rare earth element over heavy rare earth element ($LREE_N/HREE_N$) enrichment, consistent with Ca-silicate perovskite crystallization from a LREE-enriched, low-degree carbonated melt.

## Sublithospheric diamond ages

Diamond J1 from Collier-4 contained an approximately 100 μm eclogitic (that is, low in nickel) pyrrhotite in its core (Supplementary Tables 5–7). Its highly radiogenic $^{187}Os/^{188}Os$ ratio ($8.61 \pm 1.12$), coupled with a probable radiogenic initial ratio of a subducted oceanic crust precursor (Supplementary Table 8 and Fig. 1a), results in a Re–Os mantle model age of 548 to 609 million years (Ma), depending on assumptions made about the composition of the mantle and the nature of the reacting subduction fluids. The same diamond (J1) contained a breyite inclusion in its rim, with a $^{238}U/^{206}Pb$ age of $101 \pm 7$ Ma (ref. 11). The younger age probably reflects equilibration at the time of ascent (kimberlite eruption age, $93.1 \pm 1.5$ Ma (ref. 23)) rather than renewed diamond formation. The age is complicated by secondary-ion mass spectrometry sampling of only a portion of an unmixed Ca-Ti silicate perovskite inclusions which exsolved into different phases (such as breyite, larnite or titanite) with varied U/Pb upon retrogression[15].

The Juína Ca-silicate inclusions can be divided into three groups based on normalized rare earth element patterns ($REE_N$; Fig. 1b inset): group A with positive Ce anomalies, steep $Sm_N/Eu_N$ and relatively flat heavy $REE_N$; group B with flat light $REE_N$ and heavy $REE_N$ patterns; and group C with limited Ce anomalies and light $REE_N$ over heavy $REE_N$ enrichment. Group A inclusions have trace element patterns resembling oceanic crustal components that have experienced interaction with seawater. These inclusions define an Rb–Sr isochron age of $389 \pm 114$ Ma ($n = 3$; Fig. 1b) with a high $^{87}Sr/^{86}Sr$ initial ratio ($0.70909 \pm 0.00022$) that is

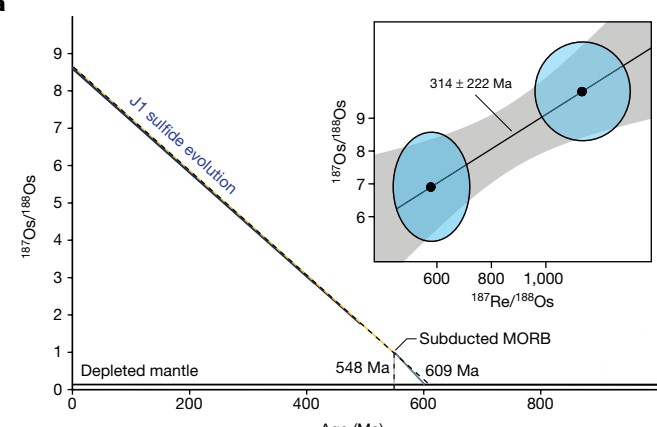
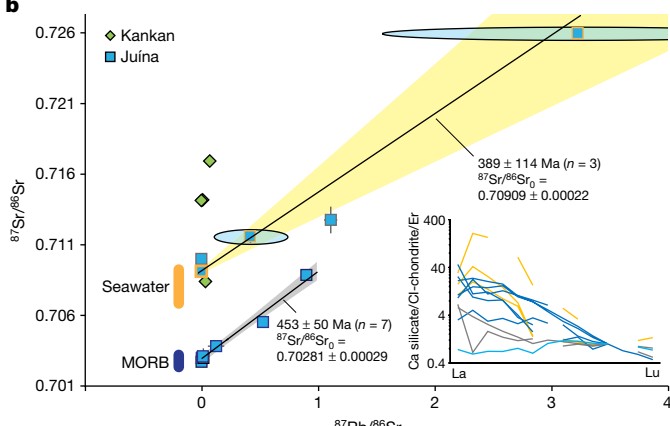
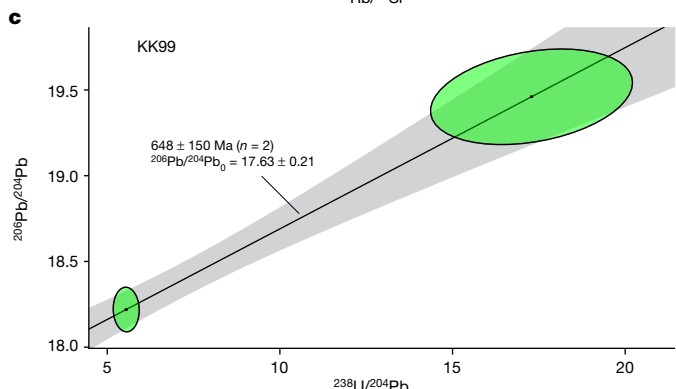

**Fig. 1 | Isochron and model ages of Juína and Kankan sublithospheric inclusions. a**, Re–Os isotope evolution of sulfide inclusion J1. It is assumed that a period of 1.5 to 3.5 Myr is required for subducting slabs to reach depths of 300–700 km (depth of diamond formation) at the ultrafast spreading rates (20 cm per yr) common in the Palaeozoic[39]. The inset shows the Re–Os equilibration age (95% confidence), as these Re–Os systematics come from two fragments of the inclusion J1 and show an age within error of the kimberlite eruption age (93 Myr, ref. 23). **b**, Rb–Sr isotope systematics (95% confidence) of Ca-silicate inclusions and micro-inclusion areas, with the maximum likelihood with overdispersion approach for isochrons in IsoplotR (model 3). The inset shows the three different REE groups (group A in yellow, group B in grey and group C in blue) of Juína Ca-silicate inclusions[30], normalized to CI-chondrite and to erbium, that were used for isochron grouping. For group C, the six inclusions with similar trace element patterns provide an age of 359 Ma. Including all inclusions ($n = 7$) increases the age to 453 Ma. Hence, a younger group cannot be excluded. The Sr isotope variation of seawater (orange vertical bar) is shown for 0–600 Ma (ref. 24) and global MORB (blue vertical bar) for $^{87}Rb/^{86}Sr$ ratios of less than 0.15 are from PetDB Database (http://www.earthchem.org/petdb). **c**, U–Pb isotope systematics (95% confidence) of micro-inclusions in diamond KK99 and its Ca-silicate inclusion, with an age of $648 \pm 150$ Ma, indicating the last time of equilibration during diamond formation.

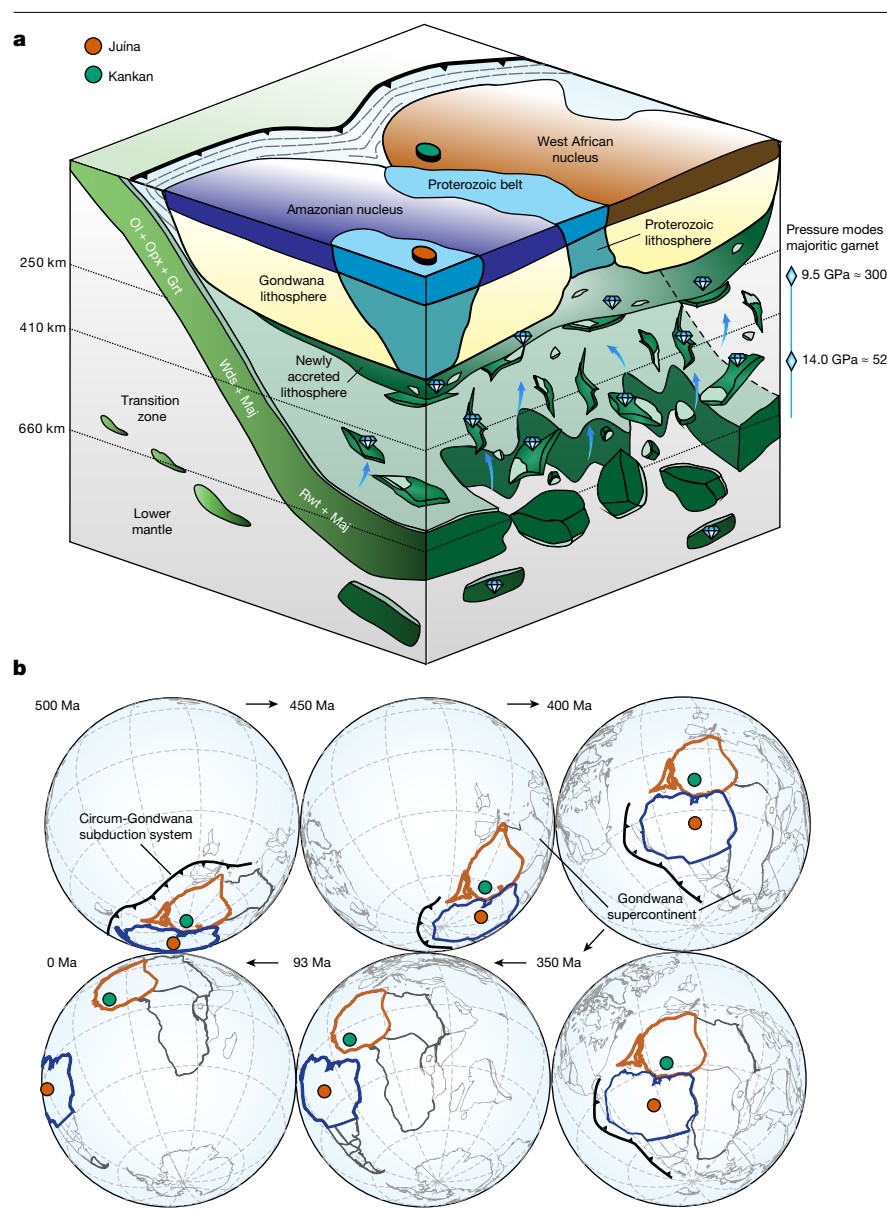

**a**

Juína
Kankan

West African nucleus

Proterozoic belt

Amazonian nucleus

Proterozoic lithosphere

Gondwana lithosphere

250 km

Newly accreted lithosphere

410 km

Transition zone

660 km

Lower mantle

Ol + Opx + Grt

Wds + Maj

Rwt + Maj

Pressure modes majoritic garnet

9.5 GPa ≈ 300 km

14.0 GPa ≈ 520 km

**b**

500 Ma → 450 Ma → 400 Ma

Circum-Gondwana subduction system

Gondwana supercontinent

0 Ma ← 93 Ma ← 350 Ma

**Fig. 2 | Sublithospheric diamond formation and continent accretion followed by supercontinent dispersal. a**, Synoptic cartoon showing sublithospheric diamond formation during Palaeozoic subduction beneath the Amazonian and West African cratonic nuclei and the subsequent diapirism of parts of the deeply subducted slab before migration and dispersal of the Gondwana supercontinent. Blue arrows show vertical accretion of new continental lithosphere (green layer) beneath Archaean and Proterozoic lithospheric roots. Peaks and ranges in equilibration depths of majoritic garnets from sublithospheric diamonds are shown as blue diamonds plus bars (Fig. 3). Slab mantle lithosphere mineralogy shown: olivine, Ol; orthopyroxene, Opx; garnet, Grt; wadsleyite, Wds; majorite, Maj; ringwoodite, Rwt. **b**, Plate reconstruction highlighting the relative juxtaposition of the Amazonian (blue) and West African (brown) portions of Gondwana and post-Gondwana regions of Earth. Note the significant northward movement between 450 Ma and 400 Ma preceding Gondwana breakup. The temporal sequence of diagrams are three-dimensional orthographic projections made with GPlates[41] and the EarthByte Global Rotation model[42–45]. Confirmed subduction zone locations are shown[32].

within the range of seawater compositions through the Phanerozoic[24]. Group B comprises only two inclusions and no isochronous relationships are interpreted here, but based on variable Ce anomalies and flat light $REE_N$ and heavy $REE_N$ patterns, their protoliths interacted with seawater. Group C inclusions have trace element patterns consistent with crystallization from low-degree melts from oceanic crust/lithosphere, an origin also supported by a low initial $^{87}Sr/^{86}Sr$ ratio (0.70281 ± 0.00029) comparable to Phanerozoic oceanic crust (mid-ocean ridge basalt (MORB); Fig. 1b) and the presence of low-Ti and high-Ti Ca-silicates. These inclusions yield a Rb–Sr isochron age of 453 ± 50 Ma ($n = 7$). Further, two published analyses of tens of combined majoritic garnet inclusions from Juína with low Rb content have Sr initial isotope ratios (0.7024–0.7032) (ref. 12) comparable to our Juína Ca-silicates from group C. Neoproterozoic or younger ages are further supported by inclusions with significant Rb ($^{87}Rb/^{86}Sr > 0.5$) that must have formed in less than 690 Myr (Extended Data Fig. 9) to maintain initial $^{87}Sr/^{86}Sr$ ratios equal to or higher than the mantle (approximately 0.702). The distinct initial $^{87}Sr/^{86}Sr$ ratios for the two isochron arrays, as well as the presence of low-Ti and high-Ti Ca-silicates, indicate that their parental melts were sourced from different portions of a subducted oceanic lithosphere.

The same inclusions were analysed for Sm–Nd and U–Pb isotope systematics. The Juína Sm–Nd data define no isochronous relationships but

imply an age of less than 530 Ma (Extended Data Fig. 10). Group C Juína Ca-silicates define a $^{206}Pb/^{204}Pb–^{238}U/^{204}Pb$ correlation of 649 ± 352 Ma (Extended Data Fig. 11). The Sm–Nd and U–Pb age constraints are within error of the Rb–Sr isochron ages (389 ± 114 Ma; 453 ± 50 Ma) and slightly younger than the Re–Os model age range (548–609 Ma), displaying remarkable agreement between four isotope systems for inclusions from Juína diamonds.

The radiogenic isotope systematics of Kankan inclusions are complicated by the incorporation of older crustal components into the diamond-forming fluids. They have extremely radiogenic $^{87}Sr/^{86}Sr$ ratios (0.714–0.716), unsupported by their very low Rb/Sr ratios, along with subchondritic Nd isotope compositions. These Sr isotope ratios are above seawater compositions and, in combination with the Nd isotope compositions yielding ancient depleted mantle Sm–Nd model ages (around 2.3 to 2.9 billion years), indicate the incorporation of Palaeoproterozoic–Neoarchaean crustal components in the diamond-forming fluids (Fig. 1b and Extended Data Fig. 12). Despite these complexities, the micro-mineral/fluid inclusion-bearing ablated diamond area and macro-inclusion within diamond KK99 define a $^{206}Pb/^{204}Pb–^{238}U/^{204}Pb$ correlation of 648 ± 150 Ma (Fig. 1c) that is in agreement with the Juína Rb–Sr, Sm–Nd, U–Pb and Re–Os ages, and therefore is taken to be the age of Kankan diamond formation. Overall, the combined isotope

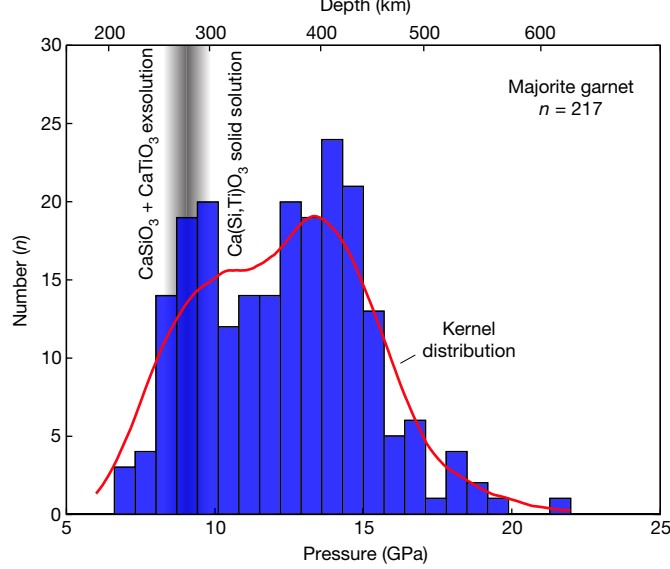

**Fig. 3 | Bimodal pressure distribution of majoritic garnet inclusions in diamonds.** The global compilation[3,46] shown here represents minimum pressures, as these are uncorrected for clinopyroxene exsolution. Including the exsolution could affect pressure estimates by up to 7 GPa (ref. 4). The mode at lower pressures indicates a residence of 7–11 GPa, approximately 210–330 km depth, at the base of the lithosphere. The presence of exsolution of $CaSiO_3$ and $CaTiO_3$ (ref. 11) indicates a portion of the sublithospheric diamonds is stored at less than approximately 9 GPa.

data show that Juína and Kankan sublithospheric diamonds formed at a similar time.

## Deep carbon cycle

Sublithospheric diamond formation ages can be used to determine residence times of recycled carbon in the mantle, tracing a deep subduction path beyond the mantle wedge (Supplementary Table 9). The Palaeozoic sublithospheric Juína and Kankan diamonds, as well as the older CLIPPIR (Cullinan-like, large, inclusion-poor, pure, irregular and resorbed[13]) and blue Type IIb sublithospheric diamonds from the 1,150-Myr-old Premier kimberlite on the Kaapvaal Craton, suggest carbon recycling[13,25,26] occurred to mantle transition zone depths since at least the Mesoproterozoic. Since all diamonds older than Mesoproterozoic have crystallized within the lithospheric mantle at depths shallower than 250 km (ref. 27), a shift to a deeper mode of carbon recycling is implied for younger times. The demonstration that sublithospheric diamonds carry surface alteration signatures in their C, N, B and Fe isotopic compositions[3,26,28,29] and record timescales comparable to convective cycling of the oceanic mantle hints at recycled material moving through the mantle and provides a mechanism to create chemical and lithological mantle heterogeneity in ocean island basalt and MORB mantle reservoirs. Moreover, from the ages for sublithospheric diamonds studied here, it could be that the circumferential subduction systems associated with supercontinents localize this recycling and consequent diamond formation.

## Diamond formation and the supercontinent cycle

An accepted model for sublithospheric diamond formation involves fluids derived from a deeply subducted slab[2,21,28,30]. The comparable time of sublithospheric diamond formation indicated by Juína and Kankan inclusions, combined with their locations in adjacent parts of the Gondwana supercontinent, suggests diamond formation events in the transition zone and lower mantle took place in a similar tectonic

setting of peri-Gondwanan subduction deep below Gondwana's lithosphere (Fig. 2a).

The temporal association of Juína and Kankan diamonds with peri-Gondwana subduction, combined with their subsequent eruption through Gondwanan lithosphere in the Cretaceous, after the dispersal of the supercontinent, implies a link between sublithospheric diamonds and the supercontinent cycle. During the late Neoproterozoic and early Cambrian, all continental plates resided in the Southern Hemisphere[31]. Thus, the bounding subduction zones to Gondwana placed subducted slabs into the deeper mantle of the Southern Hemisphere[5]. Since Kankan sublithospheric diamond formation at 648 ± 150 Ma and Juína diamond formation at around 610–450 Ma, the South American and African continents have migrated significantly. Regardless of which global plate motion model is chosen, the relative positions of South America and Africa moved rapidly north after about 450 Ma (Fig. 2b; greater than 6,500 km in latitude, approximately 60° for a Kankan reference location[31,32]). Hence, the long migration of the supercontinent and its dispersal, together with mantle convection, would be expected to move sublithospheric diamonds in the transition zone/lower mantle away from the continental mass of Gondwana. However, Kankan and Juína kimberlites, carrying sublithospheric diamonds, erupted through Gondwanan lithosphere.

We propose that the Juína and Kankan sublithospheric diamonds became accreted to the base of the Gondwana supercontinent by mantle upwelling (Fig. 2a) before long-distance migration of the supercontinent. Attachment to the keel would explain the perplexing association of sublithospheric diamonds formed in the early Cambrian to Neoproterozoic times beneath Gondwana with their appearance in Cretaceous kimberlites. The mechanics of accreting sublithospheric diamond-bearing mantle to the base of the lithosphere are unclear, but attachment of new material to the mantle root beneath Gondwana may be important in the growth and stabilization of mantle keels during supercontinent formation. While upwelling of diamond-bearing material with a mantle plume is one possibility, no evidence for a nearby plume (or plumes) between 650 and 450 Ma has been documented in a sub-Gondwana system dominated by subduction. Accretion seems more likely to be related to upwelling of depleted mantle material that gravitationally decouples from the deeply subducted slab, as originally suggested by some authors as a way for growing Archaean cratonic keels[33,34]. The depleted nature of the metaperidotitic sublithospheric diamond substrates[35] would promote buoyancy after thermal equilibration, enhancing the likelihood of vertical accretion of diamond-bearing mantle to the continental root. Modelling has previously shown that small slivers of eclogitic material may also be entrained in harzburgitic diapirs during slab disintegration and viscous overturning[36].

The diapiric uprise of depleted material and pieces of slab crust and their accretion to the lithospheric root are supported by low-Ti and high-Ti calcium perovskite phase relations and majorite inclusion geobarometry. Phase relations and minor element compositions of retrogressed former Ca- and Mg-silicate perovskite inclusions (Kankan) and Ca-silicate perovskite and majorite (Juína), indicate re-equilibration and intermediate storage of sublithospheric diamonds at a depth of 300 km or less[18,37]. Globally, a bimodal distribution of minimum pressure constraints of majoritic garnet in diamonds[3,4] also indicates the shallower mode may reflect residence at the base of the lithosphere (Fig. 3). The unmixing of Ca-silicate perovskite into $CaTiO_3$ and $CaSiO_3$[11] indicates storage at lower pressures (Fig. 3). Further support for storing sublithospheric diamonds at shallower depths for prolonged periods is provided by the presence of nitrogen in A centres (nitrogen pairs ($N_2$) rather than the more aggregated $N_4V$ in the lattice) in some sublithospheric diamonds, suggesting a potentially widespread second stage of diamond growth in the subcontinental lithospheric mantle[38].

Kankan and Juína sublithospheric diamond formation between 650 and 450 Ma, combined with the rapid northward migration of

Gondwana starting at around 450 Ma, restricts the maximum time available for diapiric rise to the base of the lithosphere to about 200 Myr. The range of typical mantle convection/plate divergence rates of 1–20 cm per year[39] allows upwelling from 660 to 250 km depth in 2–41 Myr, well within the available time constraint. We predict that other Cretaceous kimberlites with sublithospheric diamonds that erupted through former Gondwanan lithosphere, such as Letšeng (86 Ma), Koffiefontein (89 Ma), Jagersfontein (85 Ma) and Monastery (90 Ma) in the Kaapvaal Craton, could also carry evidence for such vertically accreted mantle. Indications of the ancient nature of such material have already been found at Letšeng[28]. Sublithosphere-derived mantle xenoliths from both Jagersfontein (Kaapvaal Craton) and Koidu (West African Craton, 200 km from Kankan) provide additional evidence for accretion to the roots of this former Gondwanan lithosphere before northward continent migration[40]. This model of vertical lithosphere accretion will thicken existing cratonic roots as well as welding together continents (Fig. 2a), enhancing their stability. Similar to granite plutons stabilizing cratonic crust at shallow levels, accreting lithosphere from below could stabilize the lithospheric mantle and may result in diamonds residing in Proterozoic mobile belt locations previously unexplored.

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

# Methods

Most diamonds in this study were previously studied for mineral inclusion assemblages, carbon isotope composition and nitrogen systematics of the diamond host, as indicated in Supplementary Table 1. Diamonds not previously studied were polished into plates and characterized for mineral inclusion assemblages by Raman spectroscopy and for nitrogen systematics by FTIR, as described below. Subsequently, Ca-silicate inclusions were ablated or directly digested in situ and analysed for trace elements and Rb–Sr, Sm–Nd and U–Pb isotope systematics. The sulfide was analysed by Raman, subsequently cracked out of the diamond and studied by X-ray diffraction, energy-dispersive spectroscopy and analysed for Re–Os isotope systematics. Raw data and data reduction files are provided in the EarthChem repository[47].

## Raman spectroscopy

Mineral inclusions in the diamonds were identified using a Horiba LabRam HR Evolution Raman microscope with a 532 nm green laser at the University of Alberta. Spot analyses were conducted with 20–100% laser power, a ×100 long working distance objective with a spot size of 0.7 μm, and 2–3 accumulations of 10–30 s exposures. Spectra were taken from 150 to 1,250 $cm^{-1}$ Raman shift wavelength to avoid the strong diamond Raman peak at 1,332 $cm^{-1}$. Using qwplot (a Python code written by Q. Zhang) spectra were baseline corrected and matched to spectra from the RRUFF database and superdeep mineral database[48,49]. Results are provided in Supplementary Table 3. Inclusions in diamonds Ju52, Ju104 and ColN18 were analysed on a Thermo DXRxi with a 532 nm green laser at the University of Bristol.

The sulfide was analysed at the Earth and Planets Laboratory, Carnegie Institution for Science. Confocal Raman imaging spectroscopy was undertaken using a Witec α-scanning near-field optical microscope. A frequency-doubled solid-state yttrium aluminium garnet laser operating at 532 nm was used, outputting approximately 1 mW. A ×100 long working distance objective was used and a 50 μm optical fibre acted as the confocal pinhole. Raman spectra were collected using an f/4 diafragma 300 mm focal length imaging spectrometer through a 600 lines per mm grating. A Peltier-cooled Andor electron multiplying charge-coupled device camera was attached to the spectrometer for signal collection. The scan shown in Extended Data Fig. 5 covered an area of 54 μm wide by 44 μm high, using 162 × 132 pixels, respectively, for a total of 21,384 separate spatially resolved Raman spectra. The combined image file was cosmic ray reduced, baseline corrected using SHAPE baselining and analysed for phases using Witec Project 5 software. SHAPE is an algorithm developed by Witec that uses a diameter parameter to localize the baseline correction. Assignment of Raman peaks was done with the aid of the RRUFF database[50].

## FTIR spectroscopy

Nitrogen concentrations and aggregation states of the diamonds were measured using a Thermo Nicolet Nexus 470 FTIR spectrometer in the De Beers Laboratory for Diamond Research at the University of Alberta. The FTIR spectrometer has a liquid nitrogen cooled detector attached to a Continuum infrared microscope. Transmission spectra were taken between 650 and 4,000 $cm^{-1}$ with a resolution of 4 $cm^{-1}$, an aperture spot size of 50 μm and averaged from 200 scans. Spectral deconvolution was performed using DiaMap software[51,52]. In this software, the spectrum is baselined, fitted to a Type IIa diamond spectrum, normalized to 1 cm diamond thickness, and subsequently the nitrogen peak at 1,282 $cm^{-1}$ is converted into a nitrogen concentration using an absorption coefficient of 79.4 for B centres[53]. All diamonds in this study were Type II (no observable nitrogen in FTIR; less than 10 ppm) or Type IaB (fully aggregated nitrogen, $N_4V$, in B centres). No nitrogen in A centres (nitrogen pairs, $N_2$) was detected. Results are provided in Supplementary Table 2.

## Ablation of inclusions

Unexposed Ca-silicate inclusions were ablated from the diamonds using an offline laser ablation method, following a procedure similar to the one described in McNeill et al.[54]. The diamond trace element signal is negligible (less than 1.5%) relative to the abundance of trace elements in the Ca-silicate inclusions, based on ablation tests on the same diamond that hosted the inclusions (tested for KK99, KK200 and RS-223). Diamonds were initially cleaned in Teflon-distilled concentrated $HF$-$HNO_3$ overnight at 110 °C and subsequently in 6 M HCl overnight at 110 °C, followed by rinsing with 18.2 MΩ cm Millipore water. After drying and weighing, the diamond was mounted with preleached parafilm on a plinth inside a PFA Teflon ablation cell, which was closed with a precleaned fused-silica laser window. Inclusions were ablated for several hours using a Resonetics M-50-HR laser ablation system with a 193 nm wavelength at the Arctic Resources Laboratory at the University of Alberta. Laser settings were 25–100% transmission, 91 μm size square spot, scanning speed of 100 μm $s^{-1}$, 100 Hz frequency, raster spacing of 20 μm and a fluence of 3.5–10 J $cm^{-2}$. The ablated inclusion material was collected from the cell by replacing the glass window with a Teflon cap and filling the cell with 5 ml of 6 M HCl, placing it in an ultrasonic bath for 35 min, and subsequently transferring the liquid into a 7 ml PFA vial and drying down.

## ICP-MS analyses for trace elements

Exposed inclusions (RS-122, RS-222, RS-223a and RS-223c) were digested in situ and washes of the column chemistry were collected, combined and analysed for trace elements using inductively coupled plasma mass spectrometry (ICP-MS). BHVO-2 standards that were simultaneously processed through column chemistry were compared to unprocessed BHVO-2 standards to correct for any lower trace element yield from the column. This method with BHVO-2 yield corrections has previously been proven to provide accurate trace element patterns for mineral inclusions in diamonds[55–57]. Unexposed inclusions ablated from their diamond host (RS-223b, C4-3, M-1, KK99a, KK99b, KK200a and KK200b) were dissolved in 0.5 ml of 2 M $HNO_3$, and a 20% aliquot was taken for trace element analyses before column chemistry to determine trace element patterns as well as to calculate the optimal amounts of the enriched spikes.

Samples were taken up in 3% $HNO_3$ (Optima acid, spiked with 200 ppt Rh as internal standard), for 2 h at 120 °C, and subsequently transferred to precleaned 1.5 ml centrifuge tubes. Samples, standards, total procedural blanks and centrifuge tube blanks were analysed on a Thermo Scientific Element XR2 magnetic sector ICP-MS equipped with a JET interface and Elemental Scientific Inc. (ESI) APEX-Omega unit at the University of Alberta. All elements were analysed in low mass resolution mode ($m/\Delta m$ = ~300), except for Zr and Ti which were analysed in medium mass resolution mode ($m/\Delta m$ = ~4,500). At the start of each measurement session, the ICP-MS was tuned to optimize $^{115}$In sensitivity, which was typically more than 1 million cps for a 1 ppb solution, and Ar flows were subsequently slightly decreased to reduce oxide creation. Gas flow settings are provided in Extended Data Table 1. An analysis consisted of a 180-second wash, 35-second take up time and 97-second analysis time with an uptake rate of around 200 μl $min^{-1}$. Sample counts per seconds were converted into concentrations with a five-point calibration line derived from a blank and four different dilutions (×10,000, ×50,000, ×100,000 and ×250,000) of multi-element standards from ESI. Data were subsequently corrected for total procedural blanks and weight of the sample (and in the case of the digested samples for yield loss). Results are provided in Supplementary Table 4.

For several samples an aliquot was measured by ICP-MS before column chemistry, as well as by isotope dilution for Rb–Sr–Sm–Nd–U–Pb on thermal ionization mass spectrometry (TIMS), and for these samples the trace element concentrations and ratios can be compared. All concentrations and ratios fall onto the 1:1 line and match well, except

for the smallest sample, C4-3, which is characterized by higher Rb concentrations after column chemistry (Extended Data Fig. 13). This is the sample that falls between the two isochrons when using TIMS ID concentrations and left of the two main isochrons when using ICP-MS concentrations. Rubidium blanks measured so far are consistent, but a higher blank for the column chemistry, or a much lower blank for ICP-MS work, may explain the observed difference. This sample is excluded from any age determinations.

## Column chemistry and TIMS analyses for Rb–Sr, Sm–Nd and U–Pb isotope systematics

Two Ca-silicate inclusions (Juína 3-1 and Juína 3-2) were digested in situ from their host diamond and processed for Rb–Sr isotopes at the University of Durham using the procedures outlined in Harlou et al.[58]. Rb and Sr were measured by ICP-MS using aliquots of the dissolutions. Strontium isotope ratios were analysed on a ThermoElectron Triton TIMS at the Arthur Holmes Isotope Geology Laboratory, Durham University, UK. All Sr isotope ratios were corrected using the exponential mass law to a value of 0.1194 for the $^{86}Sr/^{88}Sr$ ratio. The standard NBS987 had a long-term reproducibility of $0.710261 \pm 0.000044$ ($n = 91$; 0.6 to 0.1 ng standards) for the $^{87}Sr/^{86}Sr$ ratio, and all results have been corrected to the accepted value of 0.710250 for NBS987. A pooled blank ($n = 60$) gave an $^{87}Sr/^{86}Sr$ ratio of 0.7129 ± 0.0002 (2 s.d.). The average total procedural blank ($n = 21$) contained 5.4 ± 0.3 (2 s.d.) pg of Sr and 1.9 ± 0.7 (2 s.d.) pg of Rb, and data were corrected for those values.

The inclusions (RS-122, RS-222, RS-223a and RS-223c) were digested in situ or ablated (C4-3, M-1, RS-223b, KK99a, KK99b, KK200a, KK200b) and were processed for trace element concentrations and Rb–Sr, Sm–Nd and U–Pb isotope systematics at the University of Alberta. In situ digested samples were analysed for Rb by ICP-MS. Ablated samples were measured for Rb both by ICP-MS (see above) and subsequently by isotope dilution by TIMS. Samples were spiked with $^{87}Rb$, $^{84}Sr$, mixed $^{149}Sm–^{150}Nd$ and mixed $^{205}Pb–^{233}U–^{235}U$ (ET535) spikes. Uranium–Pb was extracted using 60 µl of AG1-X8 anion resin (Cl form, 200–400 mesh) with HBr-HCl chemistry for Fe-Ti bearing samples, with a modified procedure of Heaman and Machado[59]. The fraction containing Rb–Sr and REE was collected for further column chemistry. Strontium was separated with 50 µl of Eichrom Sr resin (100–150 um), where Rb and REE were first eluted in a total of 0.5 ml of 3 M $HNO_3$, followed by Ba elution in 0.9 ml of 7 M $HNO_3$ and finally Sr in 1.4 ml of 0.05 M $HNO_3$. The TRU spec columns for REE separation follow the procedure outlined in Koornneef et al.[60] and the LN-spec column chemistry has a set-up comparable to Pin and Zalduegui[61]. All final fractions were dried down with two to three drops of 0.05 M $H_3PO_4$ and nitrated with one to two drops of concentrated $HNO_3$ before loading for TIMS analyses.

All isotopes were analysed on a Thermo Scientific Triton Plus at the Arctic Resources Lab at the University of Alberta. Quoted standard values and blanks are averages over ten months (July 2021 to April 2022) of all diamond inclusion batches processed in the lab. Rubidium was loaded on Re filament with a $TaF_5$ activator and subsequently analysed at two different temperature blocks of 60 scans each (950–1,020 °C) with $10^{11}$ ohm amplifiers. BHVO-2 standards processed through the same column chemistry as well as ESI Rb standards yielded an average of $2.59875 \pm 0.000594$ ($n = 11$) for the same temperatures and was used as external mass fractionation correction to the natural value of 2.59265. A blank subtraction was made of 0.92 ± 0.14 pg of Rb. Strontium was analysed in static mode with $10^{11}$ ohm amplifiers on masses 86, 87, 88, and $10^{12}$ ohm amplifiers on masses 84, 85. Samples ran until exhaustion, followed by offline iterative spike and exponential mass fractionation correction to $^{88}Sr/^{86}Sr = 8.375209$ and a MAD outlier test. Sample values were normalized for the offset in NBS987 standard $^{87}Sr/^{86}Sr$ value of $0.710259 \pm 0.000011$ ($n = 11$) to the accepted value of 0.710250 and corrected for blank contributions of 8.60 ± 1.99 pg of Sr with a composition of $0.710066 \pm 0.000150$ for $^{87}Sr/^{86}Sr$.

Samarium and Nd were analysed in static mode, where the isotopes for mass fractionation correction and $^{143}Nd$ were placed on $10^{13}$ ohm amplifiers and the spiked isotope on a $10^{12}$ ohm amplifier. Like Sr, all samples ran until exhaustion, and we used an iterative approach for spike and mass fractionation correction ($^{146}Nd/^{144}Nd = 0.721902$, $^{147}Sm/^{152}Sm = 0.561134$), a MAD outlier test, and subsequent standard and blank correction. Standard values were $0.512102 \pm 0.000017$ ($n = 10$) for $^{143}Nd/^{144}Nd$ for JNdi (accepted value = 0.512115) and $0.517079 \pm 0.000034$ ($n = 9$) for $^{149}Sm/^{152}Sm$ for the Sm ESI standard. Blank amounts were 0.20 ± 0.11 pg of Nd ($^{143}Nd/^{144}Nd = 0.512034 \pm 0.000500$) and 0.034 ± 0.048 pg of Sm.

Uranium and Pb were loaded together with 4 µl silicic acid on single zone-refined Re filaments and measured with a peak-hopping method on a single ion counter, keeping a yield window of 92–93% and a dark noise of less than 1 cpm. Pb was analysed for four different temperature blocks (between 1,150 and 1,290 °C) for 50 scans each, and U was analysed for three different temperature blocks (between 1,310 and 1,360 °C) for 60 scans each, to monitor isobaric interference and large-scale mass fractionation. All data were processed using Tripoli and Redux U–Pb software[62,63] as well as the U–Pb spreadsheet developed by Schmitz and Schoene[64]. Pb mass fractionation was corrected using $^{207}Pb/^{206}Pb$ ratios of repeated analyses of the NBS981 standard and had a total mass bias (from single ion counter to Faraday to accepted value) of 0.135653% per atomic mass unit. Uranium was analysed as an oxide, using a $^{18}O/^{16}O$ oxygen isotope ratio in $UO_2$ of 0.00205 ± 0.00002. Mass fractionation was corrected using the $^{233}U/^{235}U = 0.995062$ (± 0.0054% $1\sigma$) in the ET535 tracer, and a natural $^{238}U/^{235}U$ ratio of 137.818 ± 0.045 ($2\sigma$)[65]. U and Pb blanks were 0.154 ± 0.012 pg and 1.434 ± 0.085 pg, respectively. The average Pb isotope blank ($n = 6$) has a composition of 18.94 ± 0.34% $^{206}Pb/^{204}Pb$, 15.47 ± 0.75% $^{207}Pb/^{204}Pb$ and 38.01 ± 0.64% $^{208}Pb/^{204}Pb$.

Three samples (Ju5-52, Ju5-104 (micro-inclusion area) and Col4 N-18) were processed for unspiked Sr and Nd isotope analyses at the VU University, Amsterdam. These samples were ablated at the University of Alberta. A 20% aliquot was analysed on the ICP-MS for concentrations and extrapolated to calculate 100% trace element concentrations (see above). An 80% aliquot was processed through Sr and REE chromatographic procedures similar to the ones described above. Subsequently, the Sr and REE fractions were transferred to the VU University, Amsterdam. Nd was extracted from the REE fraction with LN-spec resin columns following a procedure similar to Pin and Zalduegui[61]. Strontium and Nd were analysed on a Thermo Scientific Triton Plus at the VU University, Amsterdam. All isotope analyses were conducted in static mode, where Sr was analysed on $10^{11}$ ohm amplifiers, whereas Nd was measured on a mix of $10^{11}$ and $10^{13}$ ohm amplifiers (see Timmerman et al.[55], and Koornneef et al.[57] for more details regarding set-up and gain calibration). Analyses were corrected for exponential instrumental mass fractionation (to $^{88}Sr/^{86}Sr = 8.375209$ and $^{146}Nd/^{144}Nd = 0.721902$). Standards analysed with these samples yielded values of 0.710257 ± 0.000009 for $^{87}Sr/^{86}$ Sr (NBS987), 0.512122 ± 0.000033 (JNdi 200 ng) and 0.511331 ± 0.000039 (CIGO 0.1 ng) for $^{143}Nd/^{144}Nd$; these are within error of accepted values. Total procedural blanks were 5 pg of Sr and 0.5 pg of Nd ($^{143}Nd/^{144}Nd = 0.511752 \pm 0.005365$).

All isotopic results are provided in Supplementary Table 5 and have fully propagated errors, including uncertainties in the blank correction.

## Sulfide chemistry and Re–Os analyses

The J1 diamond plate was cracked to release the sulfide (Extended Data Fig. 4). Elemental maps produced by energy-dispersive spectroscopy revealed that the grain had oxidation at the surface (Extended Data Fig. 6). X-ray diffraction was carried out at the University of Padua to determine the crystalline structure of the bulk grain (Extended Data Fig. 7). After X-ray diffraction, the inclusion broke into two pieces. The sulfide inclusion pieces were dissolved for microchemistry procedures and analysed separately. Rhenium–Os chemistry and major element

chemistry were processed in the same batch as Zimmi sulfides and followed the same procedures described in Smit et al.[66].

### X-ray diffraction

The sulfide inclusion released from diamond J1 was analysed by X-ray diffraction in powder data collection mode, using a Supernova Rigaku Oxford Diffraction single-crystal diffractometer installed at the Department of Geosciences, University of Padua, equipped with a 200 K Pilatus Dectris detector and adopting the same experimental protocol used for sulfides as Pamato et al.[67]. The X-ray micro-source was operating at 50 kV and 0.8 mA with a beam size of 0.12 mm. The data were processed using HighScore Plus software (v.3.0e, PANalytical).

### Sulfide mineralogy, TIMS analysis for Re–Os systematics

Primary sulfides in sublithospheric diamonds are rare but have previously been found in Juína diamonds and consist of pyrite or pyrrhotite ± pentlandite[11,15,68–70]. Since most Juína diamonds are from alluvial deposits, examination of their inclusions for alteration and crystallization of secondary phases is essential. When received for this study, the sulfide in diamond J1 was exposed because a flake of diamond was missing over the sulfide. The diamond also displayed a slight rose-coloured tint near the sulfide. Based on these characteristics, the sulfide was mapped by Raman spectroscopy (see above) before diamond cracking and sulfide grain removal. The Raman mapping detected magnetite, haematite, violarite and pyrrhotite on the exposed surface of the sulfide (Extended Data Fig. 5). Molybdenite has been identified in lithospheric sulfides from the Mir kimberlite and can pose a problem for accurate Re–Os ages when Re is lost to an unrecovered molybdenite phase, potentially resulting in isochron ages older than the true age[71]. Despite molybdenite being described as a common exsolution feature of sulfides in lithospheric diamonds[71], our Raman mapping of the J1 sulfide did not detect any molybdenite. Furthermore, none of the studied sublithospheric sulfides has been shown to have molybdenite exsolution ($n = 7$; Collier-4 and Juína-5 sulfides[68]). Hence Re loss to unrecovered molybdenite is probably not an issue.

On removal of the sulfide by cracking, it broke into two pieces and scanning electron microscopy energy-dispersive spectroscopy mapping of the sulfide revealed oxygen as part of the surface composition (Extended Data Fig. 6). These energy-dispersive spectroscopy maps confirm the Raman identification of magnetite and haematite at the grain surface and necessitate careful analysis of the bulk grain by X-ray diffraction to see how much of the grain was composed of secondary phases.

X-ray diffraction (see above) of the two pieces of the extracted grains showed that its bulk mineralogy beneath the surface alteration was hexagonal pyrrhotite of the 11 M polytype with unit-cell parameters which refined to the following values: $a = 6.9071(4)$ Å, $c = 63.497(3)$ Å, $V = 2,623.4(2)$ Å$^3$ and a formula of $Fe_{10}S_{11}$. No other peaks that could be definitively correlated with magnetite, haematite, violarite or the low-temperature iron sulfide alteration phase, marcasite, were seen in the diffractogram. The absence of such peaks confirms that more than 98% of the original grain was composed of this 11 M pyrrhotite polytype. Whereas pyrrhotite is the igneous sulfide phase known to crystallize from mono-sulfide solid solution, the 11 M polytype is relatively rare in previous studies of sulfides in inclusions of lithospheric diamonds. We speculate that this rarity is because no sublithospheric sulfide inclusion has been studied previously by these methods in such detail. We also speculate that a sulfide with the formula of $Fe_{10}S_{11}$ could form by the reaction:

$$Fe_2S + Fe_3S + 4FeS_2 + FeS = Fe_{10}S_{11}$$

and thereby might confirm the natural occurrence of deep mantle iron sulfides previously experimentally predicted[72] (such as $Fe_3S$ and $Fe_2S$).

We conclude that the minor modal abundance of oxides and sulfides (less than 2%) and their position as surface alteration phases had a negligible effect on the Re–Os data reported here, which were derived from the main pyrrhotite mass of the grain. Moreover, this grain is similar to eclogitic sulfides in having both high Re and Os concentrations, which makes it less sensitive to alteration effects on Re (and Os) concentrations in a crustal environment compared to low-Re minerals such as chromite or peridotitic sulfide. Furthermore, pyrrhotite is the primary igneous Fe-sulfide to be expected in deep mantle lithologies and the two analysed pieces adhere within uncertainties to a line whose slope equals the age of kimberlite eruption (Fig. 1a inset). Thus, the Re–Os data reported here can be relied on to provide valid age constraints on the interior of Juína diamond J1.

Osmium diffusion in sulfides can help to determine whether the isotope systematics provide age information synchronous to diamond formation. Osmium diffusion rates in sulfides are fast ($10^{-13.5}$ m$^2$ s$^{-1}$ at 1,000 °C) and full isotopic resetting can occur in grains up to 500 μm at such conditions in 0.01 year (ref. 73). Therefore, even if the sublithospheric sulfide was protogenetic, Re–Os isotope equilibration should have been fast enough to reset to the time of diamond formation[67]. As both the Rb–Sr in Ca-silicates and Re–Os in sulfide record Palaeozoic ages, we can assume that the Rb–Sr system also equilibrated fast enough to yield diamond crystallization ages.

In addition to the sublithospheric sulfide studied here, one sublithospheric sulfide has previously been studied for its Re–Os isotope systematics[74]. That sulfide was recovered from an alluvial diamond from Rio Vinte e Um de Abril in the vicinity of the Aripuana-01 kimberlite pipe and had Re–Os isotope systematics elevated compared to typical fertile upper mantle[74]. As this pyrrhotite inclusion has low to intermediate Ni contents of 5–15.3 wt%, it may be considered eclogitic. Minimum and maximum ages can be constrained by the same subduction model used in our study and, with a fertile upper mantle reference reservoir, results in ages of less than 1.27 billion years ago (Ga) (ref. 74). Although the age of this sulfide is not well-constrained, its probable age (less than 1.27 Ga) is broadly consistent with the tightly constrained Phanerozoic ages of the Juína sulfide and Ca-silicate inclusions in our study.

### Inclusion and ethics

Collaboration has been established with local researchers from Brazil—co-authors I.C.N. and F.V.S.—and the outcomes are relevant for understanding superdeep diamond deposits.

### Data availability

An overview of the data is provided in the Supplementary tables. Raw and processed data is provided in Timmerman et al.[47]: EarthChem repository, https://doi.org/10.26022/IEDA/113006. Source data are provided with this paper.

### Code availability

The GPlates code was used for plate tectonic reconstructions and is open-sourced software available at https://www.gplates.org/. U–Pb data were reduced using Tripoli and Redux open-sourced software available at https://cirdles.org/. Raman peaks were identified through comparison to data from Smith et al.[45] using an in-house identification program. Other radiogenic isotope data were reduced using standard data reduction spreadsheets for spike stripping and error propagation and are provided in the EarthChem repository.

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

**Acknowledgements** We are grateful for access to the Raman laboratory of M. Steele-MacInnis. Photographs of KK99 and KK200 were shared by M. Regier. We thank De Beers Group for the donation of Kankan diamonds and Rio Tinto for Juína diamonds previously studied. We thank the Deep Carbon Observatory for funding collection and purchase of the newly studied Juína diamond (M-1) and the CPRM/SGB, Geological Survey of Brazil (Diamond Brazil Project) for the new Juína diamond (C4-3) on loan for study. To the best of our knowledge, these materials comply with responsible sampling procedures. S.T. acknowledges funding from the Government of Canada via a Banting postdoctoral fellowship. K.V.S. acknowledges GIA support for analytical visits to Carnegie. D.G.P. acknowledges funding from NSERC Discovery grant no. 418398. S.B.S. and M.J.W. acknowledge National Science Foundation grant no. EAR-2025779.

**Author contributions** S.T., S.B.S. and D.G.P. conceptualized the project and wrote the original draft. S.T. curated the data. S.T., J.M.K., R.H., G.M.N., M.Y.K., Q.Z., S.E.M.M., A.S., F.N. and K.V.S. did the formal analysis. All the authors contributed to the writing, review and editing of the final draft. T.S., D.G.P., G.B., C.B.S., S.B.S., F.K., D.Z., A.R.T., A.D.B., S.C.K., I.C.N., F.V.S. and J.W.H. carried out sample collection and/or selection.

**Competing interests** The authors declare no competing interests.

**Additional information**
**Correspondence and requests for materials** should be addressed to Suzette Timmerman.

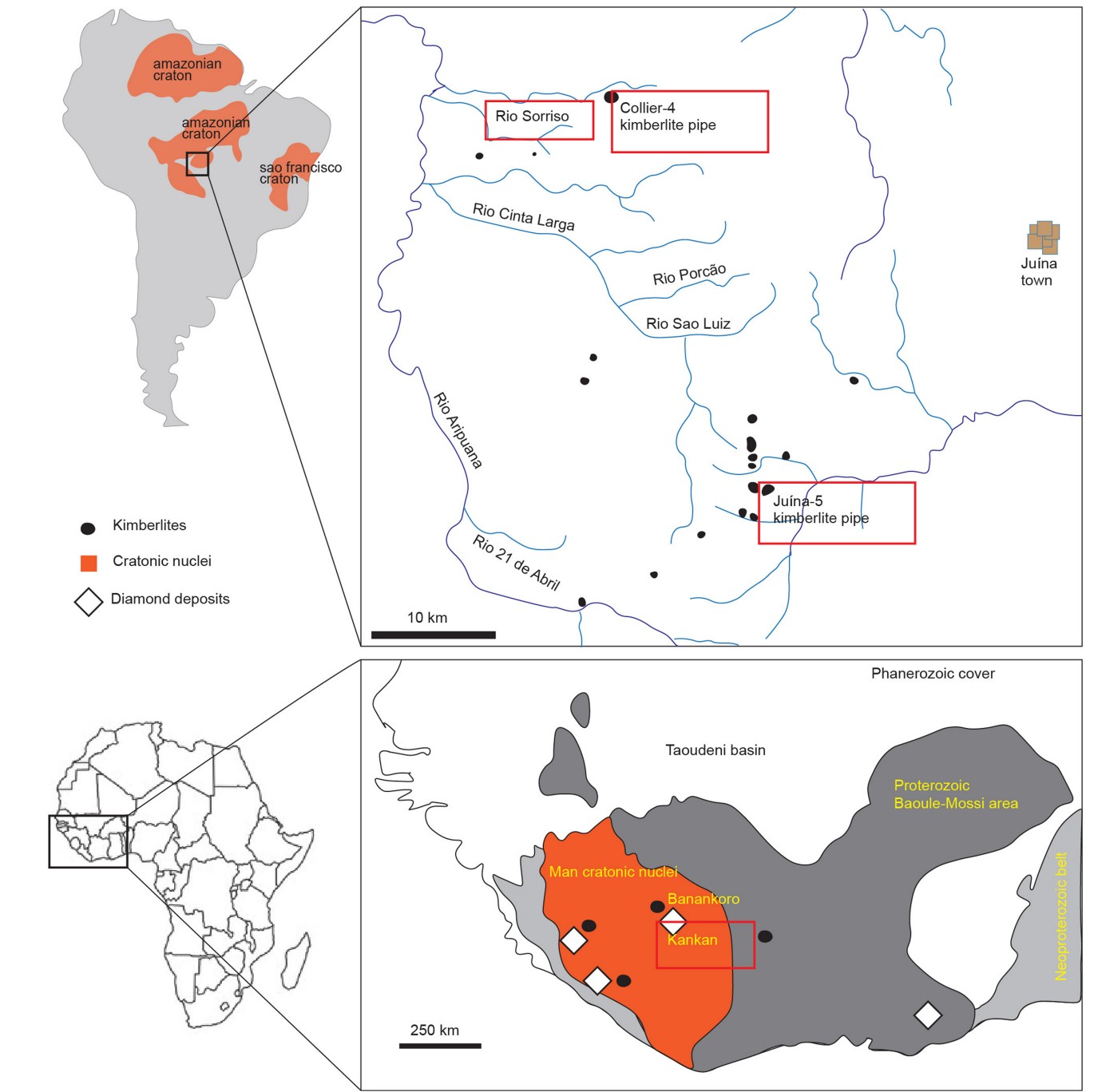

**Extended Data Fig. 1 | Sample locations in Brazil and Guinea.** Maps adapted with permission from Araujo et al.[75], Springer and Smit et al.[66], Elsevier.

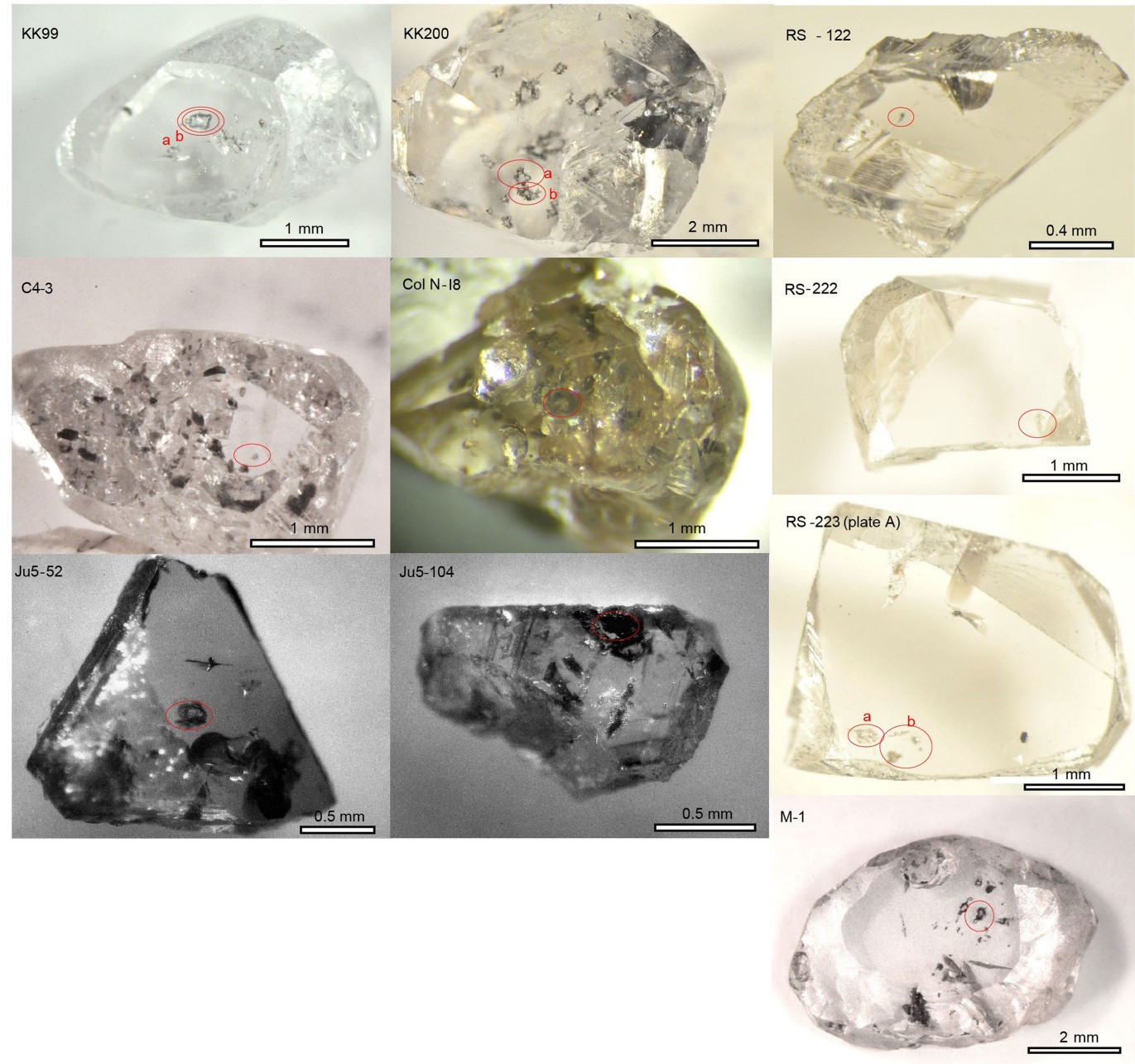

**Extended Data Fig. 2 | Overview of the studied diamonds under reflected light.** No photographs are available of Rio Sorriso alluvial diamonds Juína 3-1 and Juína 3-2. Plate C with inclusion RS-223c of RS-223 is not shown. Red ovals highlight the analysed Ca-silicate and micro-mineral/fluid inclusion-bearing areas (Ju5-104 and KK99a). As photographs were taken on different days, stage heights, and different microscropes at the University of Alberta and University of Bristol, the light colours are not directly comparable.

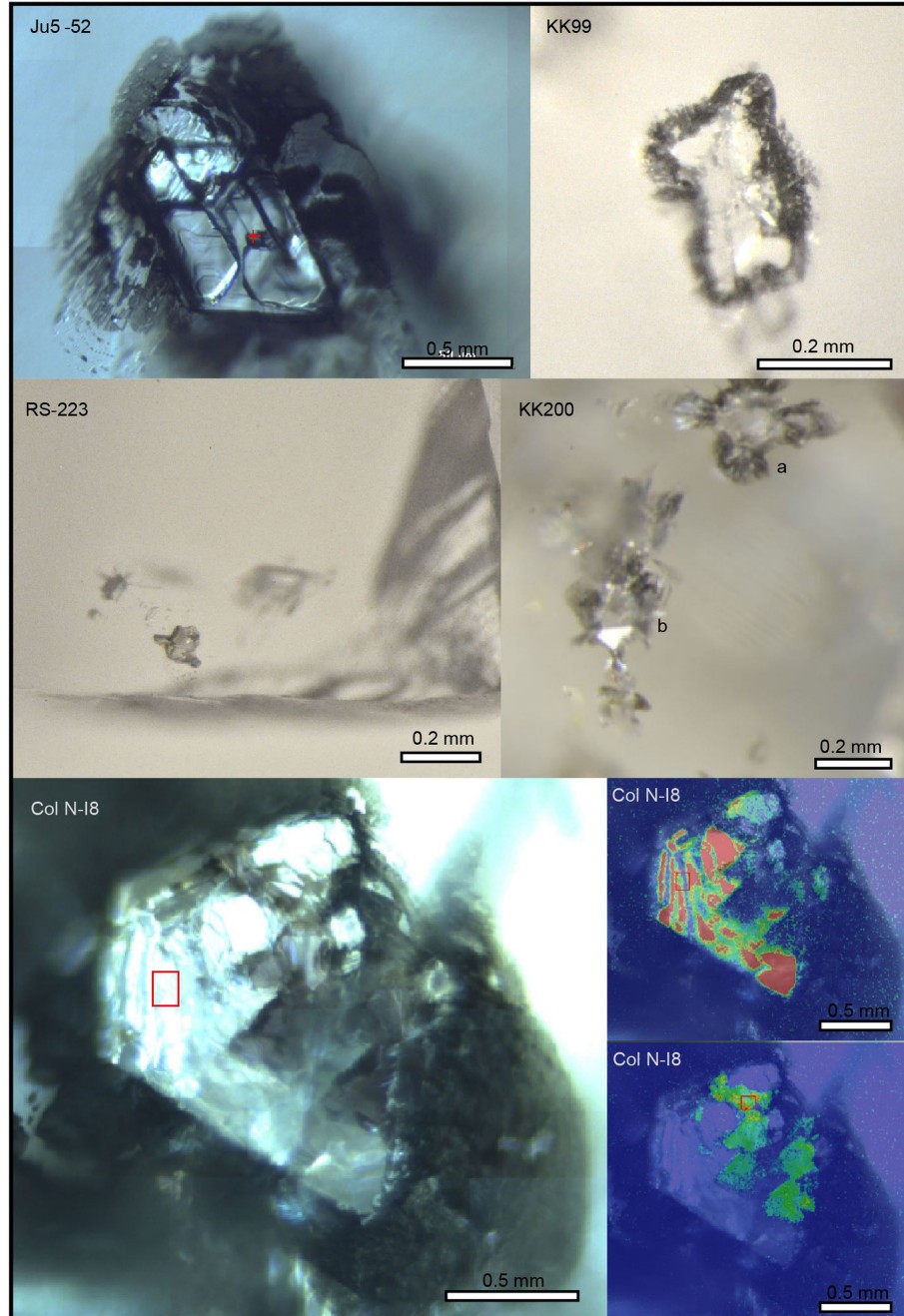

**Extended Data Fig. 3 | Examples of the studied Ca-silicate inclusions.** For the Ca(Si,Ti)$O_3$ inclusion in Col N-18, Raman mapping highlights the breyite and CaTi$O_3$ phases.

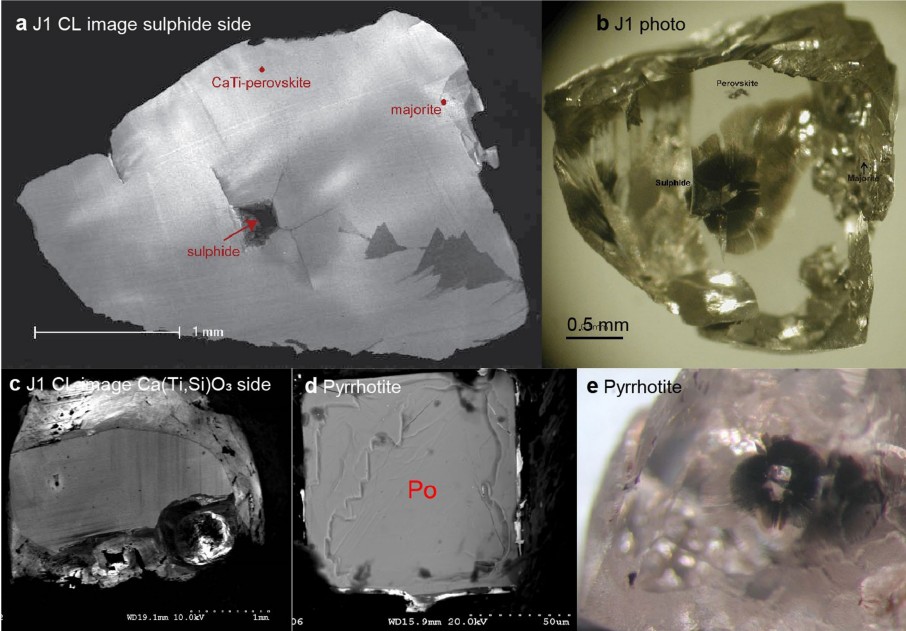

**Extended Data Fig. 4 | Images of diamond J1 and its sulfide inclusion.** The internal part of the sulfide is mostly pyrrhotite, whereas the surface of the sulfide also contained thin films of haematite, magnetite, and violarite (Extended Data Fig. 5).

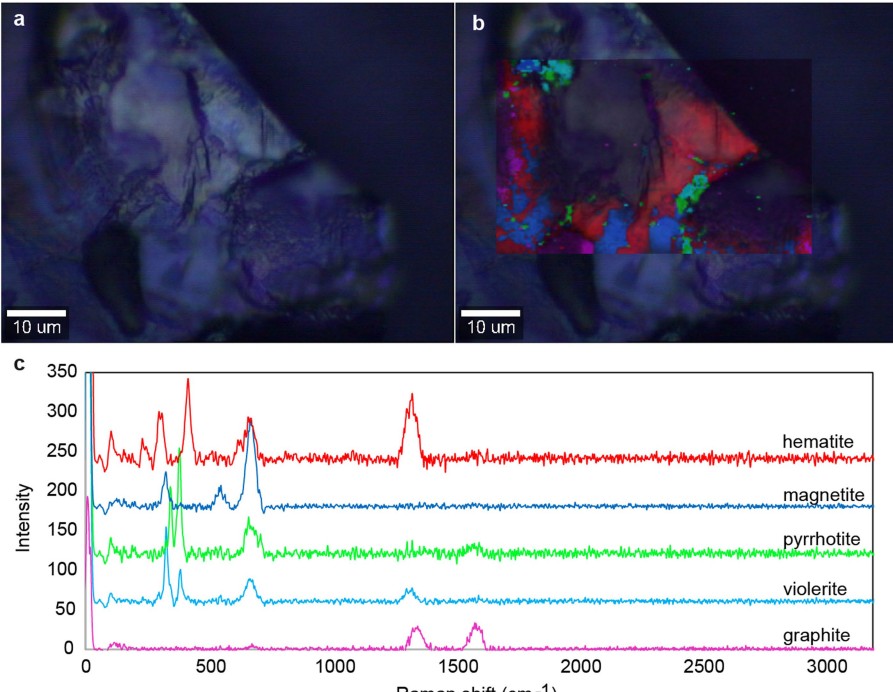

**Extended Data Fig. 5 | Raman images and spectra of J1 sulfide inclusion.** a) Reflected light image of the inclusion. b) Reflected light image of inclusion with an overlay of different phases shown by confocal Raman imaging spectroscopy. c) Spectra of phases found in the inclusion, colour-coded to match the map shown in b.

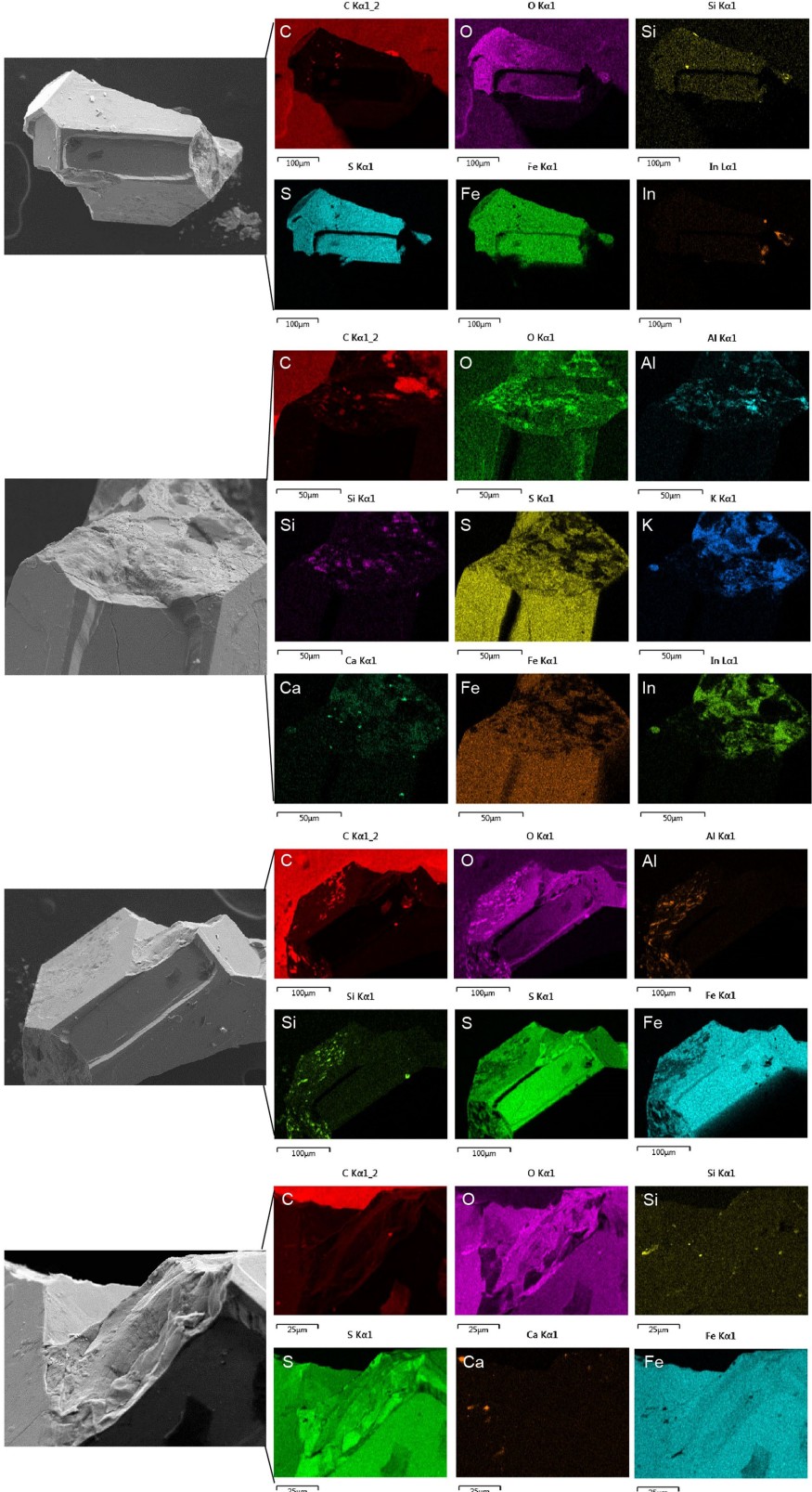

**Extended Data Fig. 6 | Elemental maps of the J1 sulfide inclusion produced by energy-dispersive spectroscopy (EDS).** Each elemental map shows the presence of a specific element in the sulfide. The presence of an element is represented in colour, black represents the lack of the element in that location in the sulfide.

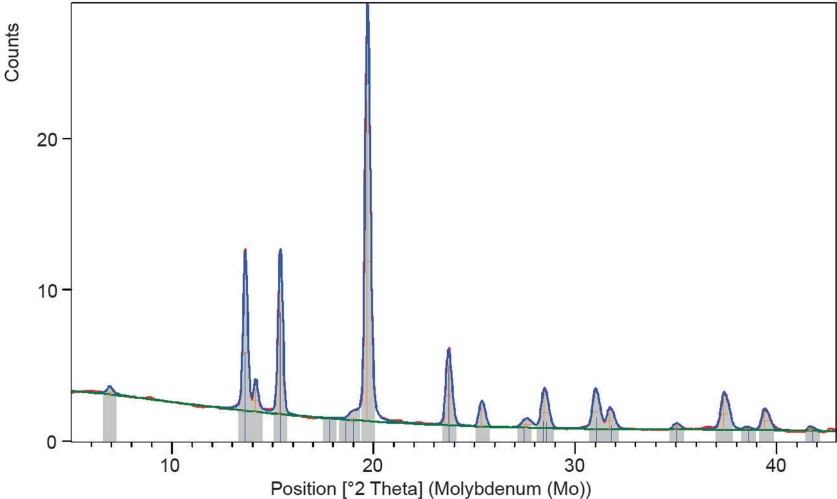

**Extended Data Fig. 7 | Micro-X-ray diffractogram of J1 sulfide inclusion.** The grey bands and the vertical blue lines correspond to the reference pyrrhotite diffractogram retrieved from the HighScore Plus software (PANalytical) database.

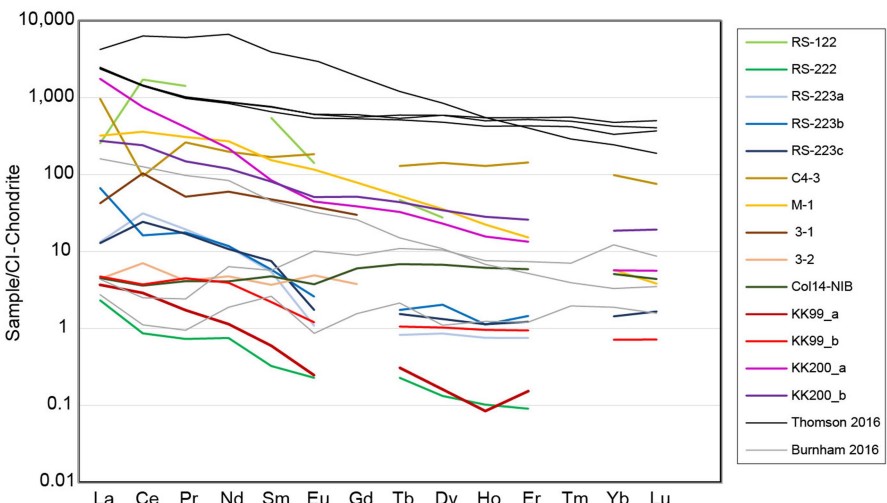

**Extended Data Fig. 8 | Rare-earth element (REE) patterns of Ca-silicate inclusions.** Normalised to CI-chondrite abundances of McDonough and Sun[76]. REE of Ca-silicates in sublithospheric diamonds from Thomson et al.[30] and Burnham et al.[77] are provided as reference here.

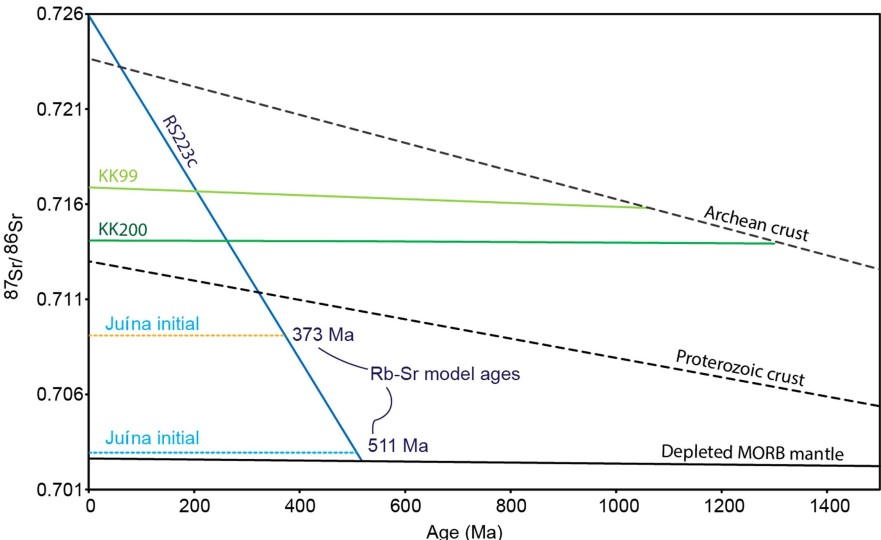

**Extended Data Fig. 9 | Rb-Sr model ages of Ca-silicate inclusions.** Juína sample RS223c has a Rb-Sr model age of 373 Ma, comparable to the isochron age of 389 Ma, using an initial ratio of 0.7091, within the range of Phanerozoic seawater compositions. A depleted mantle-like initial of 0.7028 would increase the model age to 511 Ma. Kankan samples have low Rb/Sr ratios that do not support the radiogenic $^{87}Sr/^{86}Sr$ ratios and indicate their source likely includes an older recycled crustal component. These samples were formed at <1300 Ma from an older continental crust component. Evolution curves for the Proterozoic and Archaean crust are from Banner[78], and DMM is from Workman and Hart[79]. The subduction of old continental crustal material can occur at any time after the erosion of old crust and the time that a fluid is derived from such a source can also occur at any time during or after subduction. As isotopic equilibration between any potential protogenetic inclusions and diamond-forming fluid is fast, the initial isotope compositions do not point to a protogenetic origin but rather reflect the (source of the) diamond-forming fluid/melt.

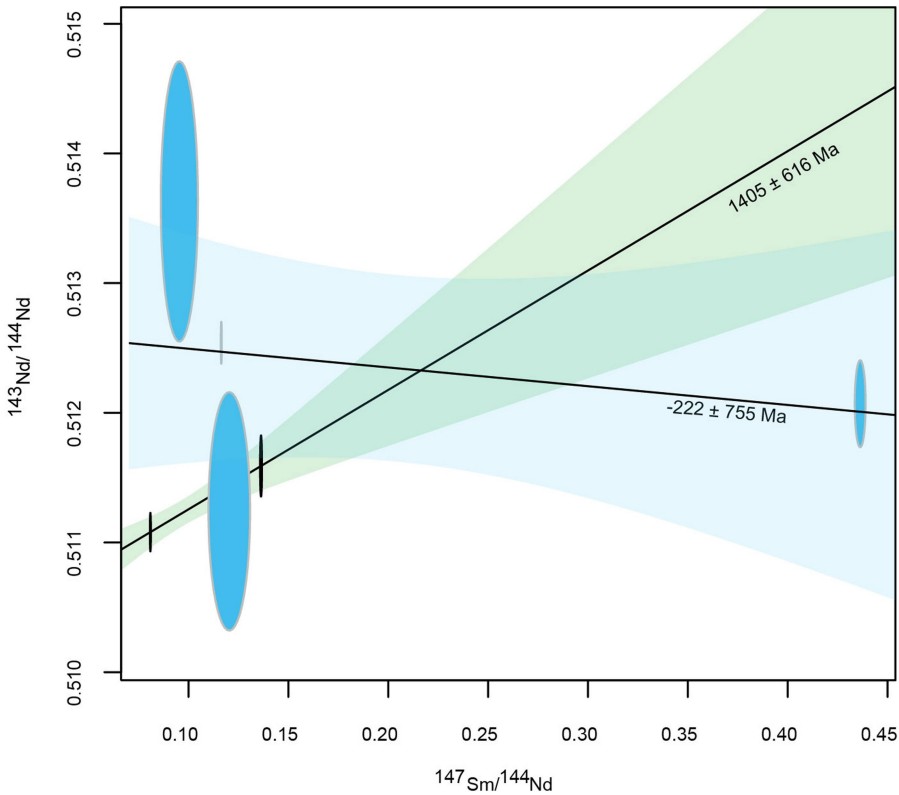

**Extended Data Fig. 10 | $^{143}$Nd/$^{144}$Nd vs $^{147}$Sm/$^{144}$Nd diagram of Juína and Kankan Ca-silicate inclusions.** Sm-Nd isotope systematics (95% confidence) did not reveal any clear isochronous relations. For Juína, Group A only had one sample with enough Sm-Nd to analyse. Group B has two samples with Sm-Nd results, but the samples have comparable Sm/Nd ratios and sample C4-3 had a large $^{143}$Nd/$^{144}$Nd error and therefore does not yield clear age information. Group C has a larger spread in Sm/Nd ratios, but there is no isochron relation, yielding average or initial $^{143}$Nd/$^{144}$Nd ratios similar to CHUR (0.51263 ± 0.00122) and indicating that if the Sm-Nd isotope systematics are equilibrated during diamond formation, the Ca-silicates (re-)crystallized <530 Ma, consistent with its Rb-Sr age of 453 Ma. For Kankan, Sm-Nd model ages using CHUR (2046–2663 Ma) and Depleted Mantle (2277–2859 Ma) reference reservoirs support an inherited older Palaeoproterozoic-Archaean component. Two Ca-silicate inclusions (from the same diamond KK200) have a two-point tie-line in Sm-Nd isotopic space with a slope equivalent to an age of 1405 ± 616 Ma with an initial $^{143}$Nd/$^{144}$Nd of 0.51034 ± 0.00047 and spread in $^{147}$Sm/$^{144}$Nd of 0.081–0.136. Given that any two points can be joined by a line, the meaning of this 'age' is very unclear and may be without any geological meaning. The two Kankan Ca-silicate inclusions were located adjacent to each other in the diamond. As no diffusion coefficients for Ca-silicates at high pressures exist, it cannot be excluded that Sm-Nd isotopes did not equilibrate sufficiently fast, thus recording a mix between the old source component and Palaeozoic diamond crystallization. The U-Pb systematics of the two KK200 inclusions records a likely kimberlite eruption age of 73 ± 11 Ma, indicating U-Pb equilibration has taken place.

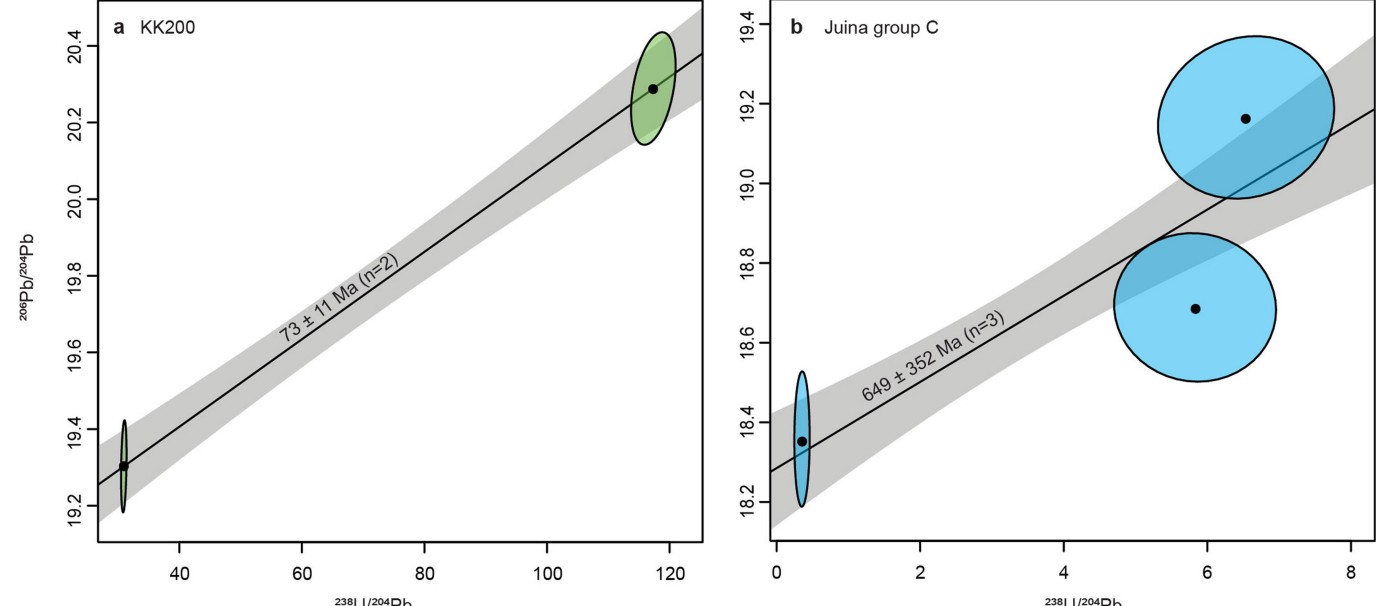

**Extended Data Fig. 11 | Traditional U-Pb isochron diagram of $^{206}$Pb/$^{204}$Pb versus $^{238}$U/$^{204}$Pb.** a) The Ca-silicate inclusions in individual diamond KK200 form an array with an age of 73 ± 11 Ma (95% confidence) that is also Cretaceous in age like kimberlites in the area, though the Banankoro kimberlites are significantly older[80]. These U-Pb systematics are interpreted to represent equilibration during kimberlite eruption between the two inclusions in KK200 as they are directly adjacent. b) Juína Group C is the only 'co-genetic' group with three sample data points for U-Pb. The three samples from Group C form a correlation with an age of 649 ± 352 Ma (2σ), which is consistent with the 453 ± 50 Ma Rb-Sr isochron age and <530 Ma Sm-Nd age of this group.

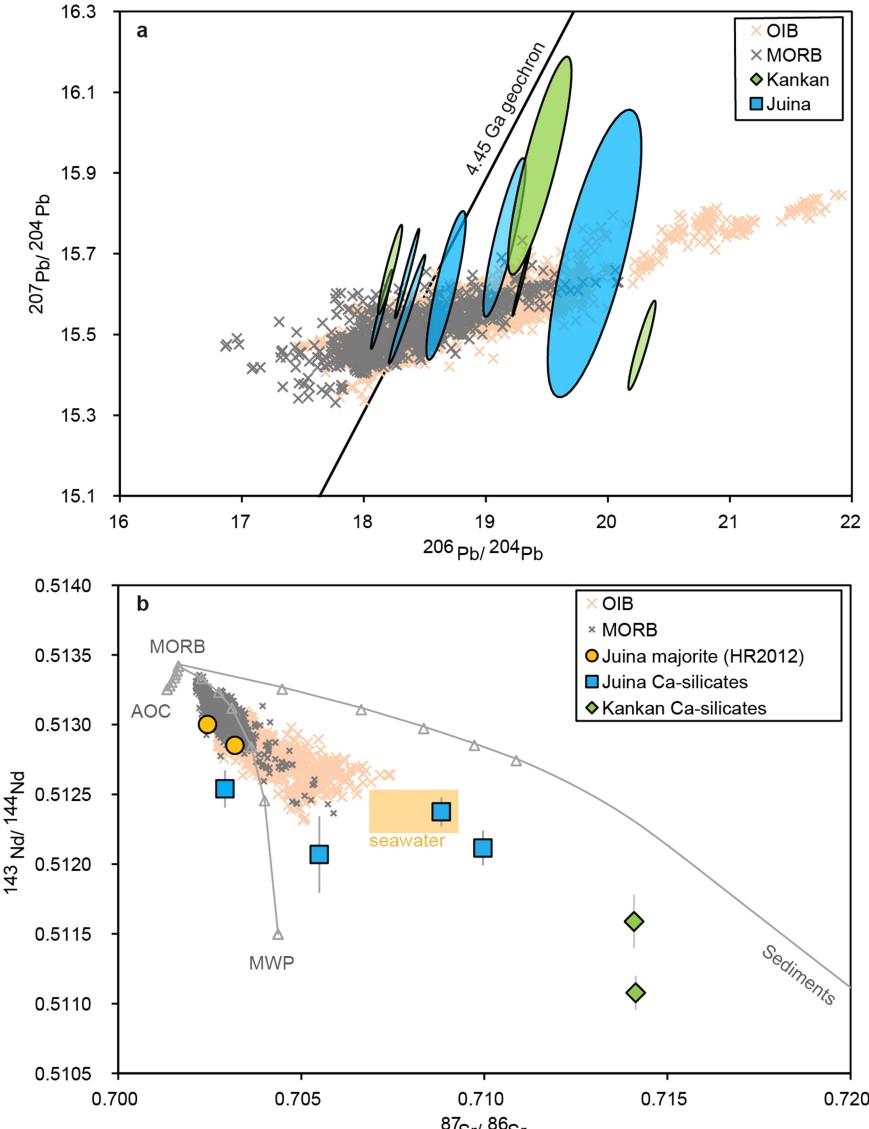

**Extended Data Fig. 12 | Pb-Pb and Nd-Sr isotope compositions of the Ca-silicate inclusions.** A) Pb-Pb isotope systematics (95% confidence) of Ca-silicate inclusions, compared to compositional ranges for present-day MORB and OIB (compilation from Stracke et al.[81]). B) Nd-Sr compositions (95% confidence) of Ca-silicates in this study and published majorite composites[12] relative to a present-day MORB and OIB compilation. Mixing components of slab and mantle are plotted, using the model of Kimura et al.[82] at 6 GPa for sediments that were formed >3.2 Ga and MORB at 3.1 Ga with subduction at 3.0 Ga. Using only DMM, or young sediments and MORB compositions cannot explain the Kankan data nor the unradiogenic Nd of Juína inclusions. Increasing pressure to conditions typical for sublithospheric diamond-forming fluids (10–25 GPa) likely results in more extreme parent/daughter ratios during melting and mixing of recycled components and thus creates larger isotopic compositional variations over time than the variation displayed in grey lines at 6 GPa here. AOC = altered oceanic crust, MWP = mantle wedge peridotite, Sed = sediments, MORB = mid-ocean ridge basalt. Triangle symbols mark the 0, 2, 4, 6, 8, 10, 20, and 100% mixing.

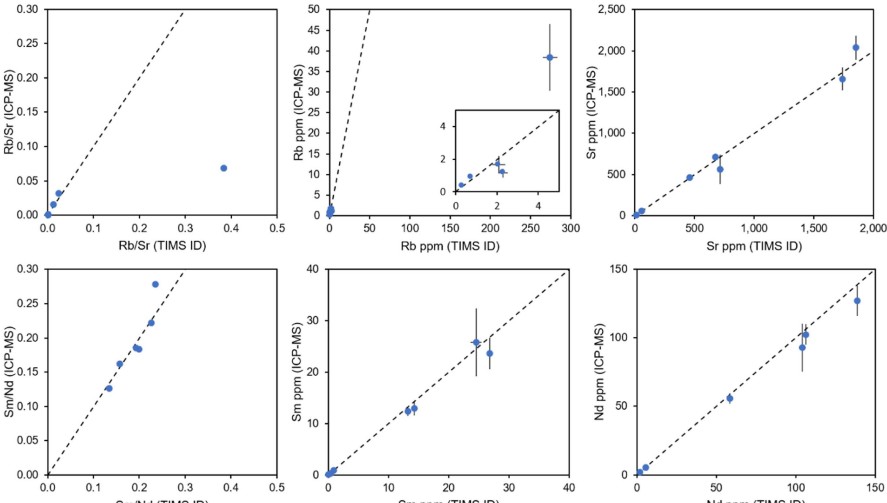

**Extended Data Fig. 13 | Comparison between ICP-MS and TIMS ID concentrations for Ca-silicate inclusions with 2 SD error bars.** In blue symbols the Ca-silicate inclusion data is shown for cases where both ICP-MS and TIMS analyses were performed on aliquots of the same sample.

**Extended Data Table 1 | Overview of gas flow rates and analytical details for trace element analyses by ICP-MS**

|  | Digested samples | Ablated samples |
|---|---|---|
| Analysis date | 10 - 11 - 2021 | 16 - 03 - 2022 |
| 3% $HNO_3$ solution (ml) | 0.45 ml | 20% aliquot in 0.4 ml |
| Uo/U (%) | 0.12 | 0.45 |
| Auxiliary Gas (L/min) | 0.80 | 0.80 |
| Sample Gas (L/min) | 1.00 | 1.05 |
| Ar flow rate (L/min) | 4.10 | 3.70 |
| $N_2$ flow rate (L/min) | 1.08 | 1.21 |