## [Peer Review File · Nature]

Manuscript Title: Sublithospheric diamond ages and the supercontinent cycle

Reviewer Comments & Author Rebuttals

Reviewer Reports on the Initial Version:

Referee #1:

The manuscript focuses on dating of tiny Fe sulfide and calcium silicate inclusions hosted by 13 sub-lithospheric diamonds, brought to the surface by ~90 Ma kimberlites in central Brazil (Juina) or Guinea (Kankan). The ages of the inclusions have been derived from a variety of isotope systems (U-Pb, Rb-Sr, Sm-Nd isochrons and Re-Os) involving state-of-the-art ablation or solution analyses at UV Amsterdam, the Universities of Alberta and Durham and Carnegie Institute. Full supporting material is provided to show the results and calculations. Although the uncertainties in some the ages are quite large, a surprise result of the investigation is the antiquity of the ages reported for some of the diamond inclusions for both Kankan and Juina, i.e. 650 Ma \pm 150 Ma 440 to 610 Ma, respectively.

Previous studies using carbon isotopes etc have shown that the sublithospheric Juina and Kankan diamonds are formed of recycled carbon. Moreover, the inclusions of high pressure phases indicate a depth of formation between ~300 and 660 km. The authors calculate that at high at the high mid-ocean-ridge spreading rates that characterised the Palaeozoic, subducted material would take only a few million years to descend from the trench to such depths. A new model is required to explain where and how these diamonds might have resided prior to their entrainment by Late Cretaceous kimberlites. The geographic locations of the diamonds are important because they are from two ancient lithospheric blocks that, prior to the break-up of the Gondwana supercontinent, were adjacent to one another, i.e. the Amazonia and West African cratons. The authors suggest that disaggregated subducted material has risen through the convecting mantle and become attached to the base of the thick continental keels, and remained there during supercontinent break-up several hundred million years later. They propose that this style of recycling of C via diamond is localised to the circumference of supercontinents, and that the accretion of once deep subducted material on to the base of cratons may be an important mechanism for the growth and stabilisation of continental keels.

General comments

1. The authors suggest that the bimodal pressures calculated for majoritic diamond inclusions are evidence of some of them stalling at the base of cratonic lithosphere. Nevertheless, the mean pressures are quite high (~9 GPa; Figure 3) and equivalent to depths that are greater than most would estimate for the thickness of the cratonic mechanical boundary layer (based on seismic tomography). The authors indicate that these pressures are minimum values but what are the uncertainties on these estimates?

2. The common mechanism for recycling subducted lithosphere is via mantle plumes triggered on the margins of 'piles' of subducted material at the core mantle boundary. The model proposed by the authors (Figure 20) implies that this material does not always descend deep into the lower mantle and enters the upper mantle convective circulation system. It seems unlikely that this material will only be manifested as diamond inclusions. Indeed, the authors model (Figure 2 indicated that this should be widespread – may there is some artistic license here). There are few places, distal to cratons and regions of thermal upwellings, where recycled material (pyroxenite) is present in erupted melts and, while these locations are quite rare, it may relate to subducted basaltic material in the upper mantle convective circulation system. For example, Hole et al. (2022) have identified ~500 Ma pyroxenitic material in basalts emplaced close to the margin of the former Gondwana supercontinent (Antarctica).

Specific comments

- Line 30. Abstract – perhaps define (quantify) what you mean by ‘deep’ to help the reader.
- Line 43. Is it the episodic mixing in the deep mantle that is driving Earth’s supercontinent cycle (or something that is associated with this cycle?).
- Line 47. State the countries of these locations so that the general reader is aware of where these are.
- Line 80. Include the formula for breyite (as you have done for larnite etc)
- Line 86. Include a formula for merwinite
- Line 90. Are these chondrite normalised? Please state.
- Line 98. Please correct the spelling of ‘composition’.
- Line 100. The U/Pb age is very close to the age of emplacement of the host kimberlite – it might be helpful to the reader if this age was cited.
- Line 105. Strictly these are rare-earth element patterns rather than trace element patterns (which is a bit more ambiguous nomenclature – I was thinking that there might be analyses of Nb, Ta etc when I first read this).
- Lines 105 to 115. I found it difficult to cross reference the REE patterns in Figure 1B with the different Groups described in this part of the text. Perhaps define these in the figure caption?
- Line 119 and 126. ‘group’ needs to be ‘Group’ to be internally consistent with the previous paragraph.
- Line 129. Change ‘close to’ to read ‘slightly lower than’.
- Line 154. Perhaps modify ‘way’ to read ‘mechanism’
- Line 180. Semantics, but the diamonds do not ‘erupt’ in the conventional use of this word, it’s the host magmas (kimberlites) that do so.
- Line 193. ‘diapiric rise’ seems rather vague
- References 62, 65 and 66. Some missing information here - please provide more details (e.g. thesis title, no of pages etc)
- Figure 1B inset. What is the normalisation that has been used here? It looks strange on the axis label. Please clarify.
- Table S8. Please define ‘O’-chondrite, i.e. ‘ordinary’
- Supplementary Files. Please present information on standards run at the same time as the inclusions.
- Line 462. Supplementary materials. I only had access to 9 Supplementary Tables (not 10 as listed). There seems to be some error with the listing of these - please check the Supplementary Tables in the .xls file (for example the tab states Table S9 but the actual Table heading states ‘Table S10: Carbon residence time calculation’).

Summary

The findings of the work are novel and based on a suite of high-precision isotope analyses on sub-lithospheric diamond inclusions. The authors acknowledge that there are outstanding questions related to their proposed model and how the newly accreted lithosphere stabilises. I’m not able to offer a better alternative to their model: while the accreted material is remobilised by mantle plume activity (and kimberlite emplacement) it remains unclear as to what stabilises it at the base of thick cratonic roots and allows it to remain isolated from convective circulation. At the very least, it must have a low viscosity (be dehydrated) and a low density. I believe that the findings in the paper will stimulate new research on mantle petrology and geodynamic mechanisms. As such it will be of interest to a wide audience and I recommend publication, subject to minor revision, by Nature.

Sally A Gibson

Referee #2:

The article presents new data on super-deep diamonds from the South America and West African Cratons. The authors show with this new data that the formation ages of these diamonds are

between 440 and 650 Ma. The authors make the point that these diamonds formed at depths of around 500 km in the transition zone and above ancient subducted slabs that surrounded the Gondwana supercontinent. The fact that Gondwana was located south of the equator 500 Ma ago when the diamonds formed and that it is now located in the northern hemisphere leads the authors to argue that after diamond formation the diamonds were transported from the transition zone up to the sublithospheric mantle and then migrated with the lithosphere to the North before continental breakup into present day South America and Africa. This proposed process implies that mantle diapirism plays an important role in the growth of the roots of continents. It further provides insights into the processes of continent migration and preservation of mantle heterogeneities.

The work is of extremely high quality and the novelty is the ability to provide consistent radiogenic isotope dates for the formation of these diamonds utilizing a variety of geochronological methods. These age constraints are combined with the identification of a subduction zone source for the diamonds. The interpretation of the data sets is well supported by the new measurements. The interpretation of the results has generally far reaching implications as it provides important evidence of the processes of continent formation.

The data is of high quality obtained at the best laboratories in the world to make these challenging measurements. The data is well presented and understandable. The methods are explained in sufficient detail and statistical approaches are included.

The conclusions are robust, significant and provide innovative new insights into the formation of the mantle roots of continents.

Suggested improvements: The main focus of the paper is the diamonds and their ages and formation history. The paper may benefit from further highlighting the significance of the findings in light of continent formation processes and the insights that can be gained with regards to material transfer processes from the deep earth to the subcontinent lithosphere. A hint at the processes to create mantle heterogeneities in MORB and OIB is provided in the section of the deep carbon cycle and the paper could benefit from some expansion of this aspect as space allows. The referred to Figure S12 is not particularly helpful and seems out of context.

Minor comments: L. 209: "A-centers" is not defined.

Fig 3: It would be helpful to indicate the depths on the figure and show the ranges for base of lithosphere and transition zone. This would help connect the figure and argument to Figure 2 (which is very helpful).

References are appropriate.

Clarity: The paper is clear and well written. It would benefit from highlighting the significance of these novel findings for continental growth models and the development of mantle heterogeneities.

*****END*****

Author Rebuttals to Initial Comments:

Line numbers refer to the manuscript with track changes.

Referee #1:

The manuscript focuses on dating of tiny Fe sulfide and calcium silicate inclusions hosted by 13 sub-lithospheric diamonds, brought to the surface by ~90 Ma kimberlites in central Brazil (Juina) or Guinea (Kankan). The ages of the inclusions have been derived from a variety of isotope systems (U-Pb, Rb-Sr, Sm-Nd isochrons and Re-Os) involving state-of-the-art ablation or solution analyses at UV Amsterdam, the Universities of Alberta and Durham and Carnegie Institute. Full supporting material is provided to show the results and calculations. Although the uncertainties in some the ages are quite large, a surprise result of the investigation is the antiquity of the ages reported for some of the diamond inclusions for both Kankan and Juina, i.e. 650 Ma \pm 150 Ma 440 to 610 Ma, respectively.

Previous studies using carbon isotopes etc have shown that the sublithospheric Juina and Kankan diamonds are formed of recycled carbon. Moreover, the inclusions of high pressure phases indicate a depth of formation between ~300 and 660 km. The authors calculate that at high at the high mid-ocean-ridge spreading rates that characterised the Palaeozoic, subducted material would take only a few million years to descend from the trench to such depths. A new model is required to explain where and how these diamonds might have resided prior to their entrainment by Late Cretaceous kimberlites. The geographic locations of the diamonds are important because they are from two ancient lithospheric blocks that, prior to the break-up of the Gondwana supercontinent, were adjacent to one another, i.e. the Amazonia and West African cratons. The authors suggest that disaggregated subducted material has risen through the convecting mantle and become attached to the base of the thick continental keels, and remained there during supercontinent break-up several hundred million years later. They propose that this style of recycling of C via diamond is localised to the circumference of supercontinents, and that the accretion of once deep subducted material on to the base of cratons may be an important mechanism for the growth and stabilisation of continental keels.

Thank you for taking the time to review our paper and providing comments to improve and clarify our manuscript.

General comments

1. *The authors suggest that the bimodal pressures calculated for majoritic diamond inclusions are evidence of some of them stalling at the base of cratonic lithosphere. Nevertheless, the mean pressures are quite high (~9 GPa; Figure 3) and equivalent to depths that are greater than most would estimate for the thickness of the cratonic mechanical boundary layer (based on seismic tomography). The authors indicate that these pressures are minimum values but what are the uncertainties on these estimates?*

This is a good point that could have been made clearer in the original MS. To clarify this, we have added: Line 221: added 'minimum' pressure. Line 481 added: 'Including the exsolution could affect pressure estimates by up to 7 GPa (29)'.

The literature barometers show 2 peaks, at 6-9 GPa and 10-15 GPa. With the machine learning barometer from Thomson et al. 2021 the mean pressure shifts up with 1.1 to 2 GPa. The pressures without taking cpx exsolution into account are minimum pressures, and the

absolute max shift is up to 7 GPa (Thomson et al. 2021). However, cpx exsolution in majorite can be almost absent to clearly present. When present it is often (our unpublished observations) present at micron to sub-micron scale such that its compositional effects will be averaged by EPMA analyses and thus the composition will render an “accurate” pressure estimate. Practically, the only way that pressures can be compared and distributions derived, is by looking at the minimum pressures from the majoritic component. The mode of 7-11 GPa (210-330 km) and peak height at 9.5 GPa are still in the range of the lithosphere-asthenosphere boundary of up to ~300 km (e.g., Gung et al. 2003; Kind et al. 2012). This boundary is poorly observed in seismic studies and material that accretes to the lithosphere may not be easily visible in seismic tomography given its thermal equilibration to asthenosphere temperatures. Our paper should provide impetus for geophysical probes to search for the presence of this material.

2. *The common mechanism for recycling subducted lithosphere is via mantle plumes triggered on the margins of ‘piles’ of subducted material at the core mantle boundary. The model proposed by the authors (Figure 20) implies that this material does not always descend deep into the lower mantle and enters the upper mantle convective circulation system. It seems unlikely that this material will only be manifested as diamond inclusions. Indeed, the authors model (Figure 2 indicated that this should be widespread – may there is some artistic license here). There are few places, distal to cratons and regions of thermal upwellings, where recycled material (pyroxenite) is present in erupted melts and, while these locations are quite rare, it may relate to subducted basaltic material in the upper mantle convective circulation system. For example, Hole et al. (2022) have identified ~500 Ma pyroxenitic material in basalts emplaced close to the margin of the former Gondwana supercontinent (Antarctica).*

This is a good point. Indeed, with our model we expect a more widespread occurrence of convecting recycled depleted lithosphere (and partial accretion to the base of continental roots) sampled by deeply derived kimberlites and likely by other magmas. It is likely that slivers of denser recycled material (recorded by high-Ti Ca silicates) can be entrained in the buoyant depleted material (presence of low-Ti Ca silicates) and convected around and sampled by basalts (as originally proposed by Ringwood, 1990). While the point about pyroxenites is interesting, there are various types of pyroxenites and numerous ways that pyroxenite can be produced in the mantle, making it very difficult to cover this issue with any satisfaction given the tight length constraints on the manuscript and the additions asked for elsewhere, but we agree that it will be an interesting line of future investigation.

Nonetheless, in an effort to allude to this issue we have added a note in line 163:

‘..mechanism to **create chemical and lithological** mantle heterogeneity in OIB and MORB mantle reservoirs...’.

We have removed the two photos inside the histogram of figure 3 and hence no artistic license is deemed necessary.

Specific comments

Line 30. Abstract – perhaps define (quantify) what you mean by ‘deep’ to help the reader.

*Agree - Done, line 33. We have reworded the sentence in the abstract to clarify: ‘the release of melts from **subducting** oceanic lithosphere at 300-700 km depths’*

Line 43. Is it the episodic mixing in the deep mantle that is driving Earth’s supercontinent cycle (or something that is associated with this cycle?).

Done, line 47. Good point, the cycle is driven by plate tectonics and episodic mixing is associated with the cycle. We have clarified this: 'Earth's supercontinent cycle, **driven by plate tectonics, results in** large-scale episodic mixing of the deep mantle due to subduction along supercontinent margins'

Line 47. State the countries of these locations so that the general reader is aware of where *these are*.

Done, line 50. We have added the countries: 'from Juina (**Brazil**) and Kankan (**Guinea, West Africa**).'

Line 80. Include the formula for breyite (as you have done for larnite etc)

Done. We have added these on line 85-86: '...breyite (CaSiO_3) inclusions, sometimes in association with larnite (Ca_2SiO_4), titanite (CaSi_2O_6)...'

Line 86. Include a formula for merwinite

Done, line 93. We have added this: 'merwinite ($\text{Ca}_3\text{Mg}[\text{SiO}_4]_2$)'

Line 90. Are these chondrite normalised? Please state.

Done, line 96. We have clarified that these are indeed chondrite normalised: '..many inclusions have **chondrite-normalised** light over heavy rare earth element ($\text{LREE}_N/\text{HREE}_N$) enrichment..'

Line 98. Please correct the spelling of 'composition'.

Done, line 104. We have fixed the spelling: 'about the **composition** of the mantle'

Line 100. The U/Pb age is very close to the age of emplacement of the host kimberlite – it might be helpful to the reader if this age was cited.

Done, line 107. We have added the kimberlite age from Heaman et al. 1998: 'The younger age probably reflects equilibration at the time of ascent (**kimberlite eruption age 93.1 ± 1.5 Ma (22)**)'

Line 105. Strictly these are rare-earth element patterns rather than trace element patterns (which is a bit more ambiguous nomenclature – I was thinking that there might be analyses of Nb, Ta etc when I first read this).

Done, line 112. We have clarified this: '..based on **rare earth** element patterns '. We do have Nb, Ta, Ba, Rb, Sr, U-Th-Pb, Y, Zr, Hf, Ti for some of the samples, see Table S4.

Lines 105 to 115. I found it difficult to cross reference the REE patterns in Figure 1B with the different Groups described in this part of the text. Perhaps define these in the figure caption?

Done, Line 447: In the figure caption we have clarified the groups: '..three different REE groups (**Group A in yellow, Group B in grey, Group C in blue**)..'

Line 119 and 126. 'group' needs to be 'Group' to be internally consistent with the previous paragraph.

Done, line 126, 135. We changed group to **Group**.

Line 129. Change 'close to' to read 'slightly lower than'.

Done, line 137. We changed 'close to' into '**slightly lower than**'.

Line 154. Perhaps modify 'way' to read 'mechanism'

Done, line 163. Changed 'way' into 'mechanism': 'provides a **mechanism**'

Line 180. Semantics, but the diamonds do not 'erupt' in the conventional use of this word, it's the host magmas (kimberlites) that do so.

Done, line 191. We clarified that we mean kimberlites erupting: 'Kankan and Juina **kimberlites, carrying** sublithospheric diamonds,'

Line 193. 'diapiric rise' seems rather vague

In the first notion of this concept we have changed it to the more general term of 'upwelling' (line 208), but we note that diapirism is in general use by the community to mean buoyancy-driven up-welling.

References 62, 65 and 66. Some missing information here - please provide more details (e.g. thesis title, no of pages etc)

We apologise for this omission – due to technical reasons. For these thesis references the details keep disappearing when the Nature referencing style is selected in Mendeley. They have been adjusted manually now.

L. I. Kemppinen, *Investigating the timing and nature of diamond-forming events through the study of diamond-hosted sulphide inclusions*. PhD thesis, 266 pages, University of Bristol, Bristol, United Kingdom (2020).

M. T. Hutchison, *Constitution of the deep transition zone and lower mantle shown by diamonds and their inclusions*. PhD thesis, 646 pages, Edinburgh, United Kingdom (1997).

The Thomson 2014 thesis citation has been changed to the Thomson et al. 2014 CMP paper, as the inclusion data we are citing is fully covered in the published paper.

Figure 1B inset. What is the normalisation that has been used here? It looks strange on the axis label. Please clarify.

The data has first been normalised to chondrite (CI-chondrite), then to Erbium. This second normalisation to Erbium (or any REE of choice) is to highlight the differences in slope, and not focus on the differences in concentration. To clarify this has been added to the figure caption: 'groups (Group A in yellow, Group B in grey, Group C in blue) of Juína Ca-silicate inclusions, normalised to CI-chondrite and to Erbium, that...'

Table S8. Please define 'O'-chondrite, i.e. 'ordinary'

As suggested we have written it as Ordinary-chondrite.

Supplementary Files. Please present information on standards run at the same time as the inclusions.

Provided: An overview of the standard averages are provided in the Method section and in the table in the online EarthChem repository. In the online EarthChem repository also an overview of the standard results and the raw files of the standard analyses that were run at the same time as the inclusions are provided.

Line 462. Supplementary materials. I only had access to 9 Supplementary Tables (not 10 as listed). There seems to be some error with the listing of these - please check the Supplementary Tables in the .xls file (for example the tab states Table S9 but the actual Table heading states 'Table S10: Carbon residence time calculation').

We apologise for the numbering error – now corrected. There are 9 tables in the excel file, and Table S10 is in the Method section embedded in the text. You are correct that Table S9 was incorrectly labelled in the heading. This has been fixed.

Summary

The findings of the work are novel and based on a suite of high-precision isotope analyses on sub-lithospheric diamond inclusions. The authors acknowledge that there are outstanding questions related to their proposed model and how the newly accreted lithosphere stabilises. I'm not able to offer a better alternative to their model: while the accreted material is remobilised by mantle plume activity (and kimberlite emplacement) it remains unclear as to what stabilises it at the base of thick cratonic roots and allows it to remain isolated from convective circulation. At the very least, it must have a low viscosity (be dehydrated) and a low density. I believe that the findings in the paper will stimulate new research on mantle petrology and geodynamic mechanisms. As such it will be of interest to a wide audience and I recommend publication, subject to minor revision, by Nature.

Sally A Gibson

Referee #2:

The article presents new data on super-deep diamonds from the South America and West African Cratons. The authors show with this new data that the formation ages of these diamonds are between 440 and 650 Ma. The authors make the point that these diamonds formed at depths of around 500 km in the transition zone and above ancient subducted slabs that surrounded the Gondwana supercontinent. The fact that Gondwana was located south of the equator 500 Ma ago when the diamonds formed and that it is now located in the northern hemisphere leads the authors to argue that after diamond formation the diamonds were transported from the transition zone up to the sublithospheric mantle and then migrated with the lithosphere to the North before continental breakup into present day South America and Africa. This proposed process implies that mantle diapirism plays an important role in the growth of the roots of continents. It further provides insights into the processes of continent migration and preservation of mantle heterogeneities.

The work is of extremely high quality and the novelty is the ability to provide consistent radiogenic isotope dates for the formation of these diamonds utilizing a variety of geochronological methods. These age constraints are combined with the identification of a subduction zone source for the diamonds. The interpretation of the data sets is well supported by the new measurements. The interpretation of the results has generally far reaching implications as it provides important evidence of the processes of continent formation.

The data is of high quality obtained at the best laboratories in the world to make these challenging measurements. The data is well presented and understandable. The methods are explained in sufficient detail and statistical approaches are included.

The conclusions are robust, significant and provide innovative new insights into the formation of the mantle roots of continents.

Thank you for taking the time to review our manuscript and suggesting improvements to highlight the implications of our data and model.

Suggested improvements: The main focus of the paper is the diamonds and their ages and formation history. The paper may benefit from further highlighting the significance of the findings in light of continent formation processes and the insights that can be gained with regards to material transfer processes from the deep earth to the subcontinent lithosphere. A hint at the processes to create mantle heterogeneities in MORB and OIB is provided in the section of the deep carbon cycle and the paper could benefit from some expansion of this aspect as space allows. The referred to Figure S12 is not particularly helpful and seems out of context.

Thank you for this excellent suggestion. We agree that we under-played the significance of this a little. We have added the following in lines 162-163: The demonstration that sublithospheric diamonds carry surface alteration signatures in their C, N, B, and Fe isotopic compositions (11, 25, 27, 28) and record timescales comparable to convective cycling of the

oceanic mantle hints at recycled material moving through the mantle and provides a mechanism to create chemical and lithological mantle heterogeneity in OIB and MORB mantle reservoirs.

We have added the following in lines 241-244: 'This model of newly accreting vertical lithosphere accretion will thicken existing cratonic roots as well as welding together continents (Fig. 2A), enhancing their stability. Similar to granite plutons stabilising cratonic crust at shallow levels, accreting lithosphere from below can stabilise the root and may result in diamonds residing in Proterozoic mobile belt locations previously unexplored.'

Also, because space is tight in the text, we have sought to increase the impact of the results on continent formation by making Figure 2 more explicit in the way that it portrays additional lithospheric growth – beneath both Archean nuclei and the Proterozoic regions of cratons. We have removed the reference to Figure S12 here.

Minor comments: L. 209: "A-centers" is not defined.

DONE - A-centers are a lattice defect of telling how the nitrogen sits in the crystal: in pairs of nitrogen. This has been clarified in line 225: 'presence of nitrogen in A-centers (nitrogen pairs (N₂) rather than the more aggregated N₄V in the lattice) in some sublithospheric diamonds'

Fig 3: It would be helpful to indicate the depths on the figure and show the ranges for base of lithosphere and transition zone. This would help connect the figure and argument to Figure 2 (which is very helpful).

AGREE - As suggested, we have indicated the 410, and 660 depths (transition zone boundaries) as well as the 250 km depth Figure 2 now and added a depth scale to figure 3. We have also indicated the majoritic peak pressures on Figure 2, to connect better to Figure 3. We think that this revised figure is much more informative now.

References are appropriate.

Clarity: The paper is clear and well written. It would benefit from highlighting the significance of these novel findings for continental growth models and the development of mantle heterogeneities.

Thank you, see the reply above regarding highlighting the significance of the findings.